# POSTERIOR-MEAN RECTIFIED FLOW: TOWARDS MINIMUM MSE PHOTO-REALISTIC IMAGE RESTORATION

**Guy Ohayon, Tomer Michaeli, Michael Elad**
Technion – Israel Institute of Technology
{ohayonguy@cs,tomer.m@ee,elad@cs}.technion.ac.il

## ABSTRACT

Photo-realistic image restoration algorithms are typically evaluated by distortion measures (*e.g.*, PSNR, SSIM) and by perceptual quality measures (*e.g.*, FID, NIQE), where the desire is to attain the lowest possible distortion without compromising on perceptual quality. To achieve this goal, current methods commonly attempt to sample from the posterior distribution, or to optimize a weighted sum of a distortion loss (*e.g.*, MSE) and a perceptual quality loss (*e.g.*, GAN). Unlike previous works, this paper is concerned specifically with the *optimal* estimator that minimizes the MSE under a constraint of perfect perceptual index, namely where the distribution of the reconstructed images is equal to that of the ground-truth ones. A recent theoretical result shows that such an estimator can be constructed by optimally transporting the posterior mean prediction (MMSE estimate) to the distribution of the ground-truth images. Inspired by this result, we introduce Posterior-Mean Rectified Flow (PMRF), a simple yet highly effective algorithm that approximates this optimal estimator. In particular, PMRF first predicts the posterior mean, and then transports the result to a high-quality image using a rectified flow model that approximates the desired optimal transport map. We investigate the theoretical utility of PMRF and demonstrate that it consistently outperforms previous methods on a variety of image restoration tasks.

## 1 INTRODUCTION

Photo-realistic image restoration (PIR) is the task of reconstructing visually appealing images from degraded measurements (*e.g.*, noisy, blurry). This is a long-standing research problem with diverse applications in mobile photography, surveillance, remote sensing, medical imaging, and more. PIR algorithms are commonly evaluated by distortion measures (*e.g.*, PSNR, SSIM (Wang et al., 2004), LPIPS (Zhang et al., 2018)), which quantify some type of discrepancy between the reconstructed images and the ground-truth ones, and by perceptual quality measures (*e.g.*, FID (Heusel et al., 2017), KID (Bińkowski et al., 2018), NIQE (Mittal et al., 2013), NIMA (Talebi & Milanfar, 2018)), which are intended to predict the extent to which the reconstructions would look natural to human observers. Since distortion and perceptual quality are typically at odds with each other (Blau & Michaeli, 2018), the core challenge in PIR is to achieve minimal distortion *without* sacrificing perceptual quality.

A common way to approach this task is through posterior sampling (Bendel et al., 2023; Chung et al., 2023; Daras et al., 2024; Kawar et al., 2021a;b; 2022; Man et al., 2023; Murata et al., 2023; Ohayon et al., 2021; Saharia et al., 2022; 2023; Song

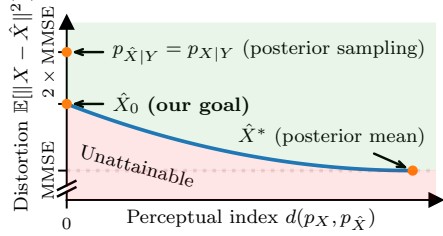

Figure 1: Illustration of the distortion-perception tradeoff, where distortion is measured by MSE. Many photo-realistic image restoration methods aim for posterior sampling. Theoretically, this approach achieves a perfect perceptual index ($p_{\hat{X}} = p_X$) but its MSE is twice the MMSE. In contrast, we aim for the estimator $\hat{X}_0$ that *minimizes the MSE* under a perfect perceptual index constraint (Eq. (3)), which typically achieves a *smaller* MSE than posterior sampling.

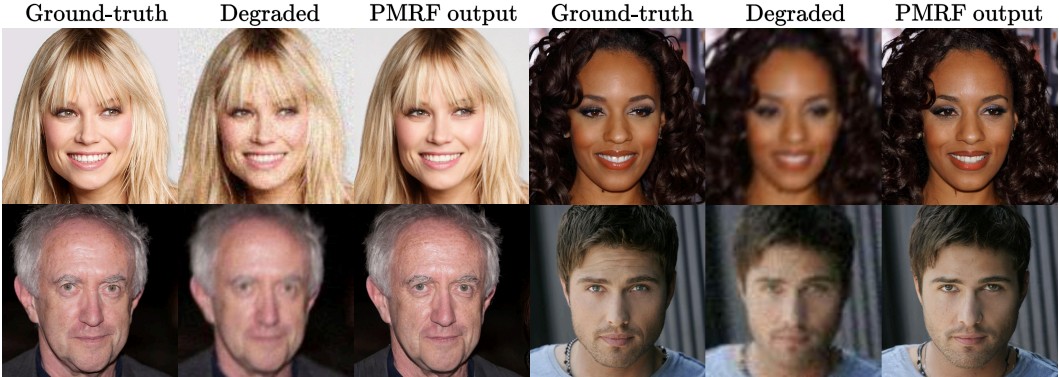

| Ground-truth | Degraded | PMRF output | Ground-truth | Degraded | PMRF output |

Figure 2: Visual results of PMRF (our method) on the **CelebA-Test** blind face image restoration data set. Our algorithm produces sharp and visually appealing details while maintaining incredibly low distortion according to a variety of measures *simultaneously*. See Table 1.

Table 1: Quantitative evaluation of state-of-the-art blind face image restoration algorithms on the **CelebA-Test** benchmark. Red, blue and green indicate the best, the second best, and the third best scores, respectively. Our method achieves the best FID, KID, PSNR and SSIM, and the second or third best scores in the rest of the perceptual quality and distortion measures. A visual comparison is provided in Figure 2 and Figure 6 in the appendix.

| Method | Perceptual Quality | | | | Distortion | | | | |
|---|---|---|---|---|---|---|---|---|---|
| | FID↓ | KID↓ | NIQE↓ | Precision↑ | PSNR↑ | SSIM↑ | LPIPS↓ | Deg↓ | LMD↓ |
| DOT | 100.2 | 0.0914 | 6.462 | 0.1600 | 21.32 | 0.6636 | 0.4756 | 43.87 | 2.876 |
| RestoreFormer++ | 41.15 | 0.0290 | 4.187 | 0.6877 | 25.31 | 0.6703 | 0.3441 | 29.63 | 2.043 |
| RestoreFormer | 42.30 | 0.0301 | 4.405 | 0.7010 | 24.62 | 0.6460 | 0.3655 | 32.13 | 2.299 |
| CodeFormer | 53.16 | 0.0425 | 4.649 | 0.6940 | 25.15 | 0.6700 | 0.3432 | 37.28 | 2.470 |
| VQFRv1 | 41.79 | 0.0297 | 3.693 | 0.6593 | 24.07 | 0.6446 | 0.3515 | 35.75 | 2.429 |
| VQFRv2 | 46.77 | 0.0346 | 4.169 | 0.6590 | 23.23 | 0.6412 | 0.3624 | 44.38 | 3.053 |
| GFPGAN | 46.72 | 0.0350 | 4.415 | 0.6970 | 24.99 | 0.6774 | 0.3643 | 36.05 | 2.443 |
| DiffBIR | 59.06 | 0.0509 | 6.084 | 0.5643 | 25.39 | 0.6536 | 0.3878 | 32.94 | 2.006 |
| DifFace | 38.43 | 0.0258 | 4.288 | 0.7413 | 24.80 | 0.6726 | 0.3999 | 45.79 | 2.965 |
| BFRffusion | 41.53 | 0.0301 | 4.966 | 0.6623 | 26.21 | 0.6917 | 0.3619 | 30.98 | 1.992 |
| **PMRF (Ours)** | 37.46 | 0.0257 | 4.118 | 0.7073 | 26.37 | 0.7073 | 0.3470 | 30.67 | 2.030 |

et al., 2023; Wang et al., 2023a; Zhu et al., 2023). Specifically, letting $X$ and $Y$ denote the random vectors corresponding to the ground-truth image and its degraded measurement, respectively, posterior sampling generates a reconstruction $\hat{X}$ by sampling from $p_{X|Y}$ (such that $p_{\hat{X}|Y} = p_{X|Y}$). This solution is appealing as it theoretically guarantees a perfect *perceptual index*[1] ($p_{\hat{X}} = p_X$). Interestingly, however, the Mean Squared Error (MSE) that this solution achieves is not the minimal possible under the perfect perceptual index constraint. Indeed, the MSE achieved by posterior sampling is precisely twice the Minimum MSE (MMSE) that can be achieved without a constraint on the perceptual index (Blau & Michaeli, 2018). This is while the minimal MSE achievable under a perfect perceptual index constraint is typically strictly smaller (Blau & Michaeli, 2018; Freirich et al., 2021), as illustrated in Figure 1. We denote by $\hat{X}_0$ the estimator that minimizes the MSE under a perfect perceptual index constraint. Its formal definition is provided in Section 2.2.

Another common way to solve PIR tasks is to train a model by minimizing a weighted sum of a distortion loss (*e.g.*, MSE) and a GAN loss (Goodfellow et al., 2014; Gu et al., 2022; Ledig et al., 2017; Wang et al.; 2018; 2021; 2022; 2023b; Yang et al., 2021; Zhang et al., 2021; Zhou

---

[1]Formally, the perceptual index of $\hat{X}$ is defined as the statistical divergence between $p_{\hat{X}}$ and $p_X$.

et al., 2022). As explained by Blau & Michaeli (2018), this is a principled way to traverse the distortion-perception tradeoff, where the GAN loss coefficient acts as a Lagrange multiplier that controls the desired perceptual index. Thus, in principle, one can approximate $\hat{X}_0$ by selecting a sufficiently large such coefficient. Despite the elegance of this approach, diffusion methods that aim for posterior sampling tend to perform better in practice, both in terms of distortion and perceptual quality (see Table 1), implying that current GAN-based methods fail to approximate $\hat{X}_0$. Such a shortcoming can be partially attributed to the fact that GANs are extremely difficult to optimize, especially when the GAN loss coefficient is significantly larger than that of the distortion loss.

In this paper, we propose *Posterior-Mean Rectified Flow* (PMRF), a straightforward framework to *directly* approximate $\hat{X}_0$. Interestingly, Freirich et al. (2021) proved that $\hat{X}_0$ can be constructed by first predicting the posterior mean $\hat{X}^* := \mathbb{E}[X|Y]$, and then optimally transporting the result to the ground-truth image distribution (see Section 2.2 for a formal explanation). Motivated by this result, PMRF first approximates the posterior mean by using a model that minimizes the MSE between the reconstructed outputs and the ground-truth images. Then, we train a rectified flow model (Liu et al., 2023) to predict the direction of the straight path between corresponding pairs of posterior mean predictions and ground-truth images. Given a degraded measurement at test time, PMRF solves an ODE using such a flow model, with the posterior mean prediction set as the initial condition. As we explain in Section 3, PMRF approximates the desired estimator $\hat{X}_0$, aiming for a solution that minimizes the MSE under a perfect perceptual index constraint.

Our paper is organized as follows. In Section 2 we provide the necessary background and set mathematical notations. In Section 3 we describe our proposed method, and provide intuition via theoretical results and a toy example with closed-form solutions. In Section 4 we discuss related work. In Section 5 we demonstrate the utility of PMRF on a variety of image restoration tasks, including denoising, super-resolution, inpainting, colorization, and blind restoration. We show that PMRF sets a new state-of-the-art on several benchmarks in the challenging blind face image restoration task, and is either on-par or outperforms previous frameworks in the rest of the tasks. Finally, in Section 6 we conclude our work and discuss its limitations.

## 2 BACKGROUND

We adopt the Bayesian perspective for solving inverse problems (Davison, 2003; Kaipio & Somersalo, 2005), where a natural image $x$ is regarded as a realization of a random vector $X$ with probability density function $p_X$. The degraded measurement $y$ (*e.g.*, a noisy or low-resolution image) is a realization of a random vector $Y$, which is related to $X$ via the conditional probability density function $p_{Y|X}$. Given a degraded measurement $y$, an image restoration algorithm generates a prediction $\hat{x}$ by sampling from $p_{\hat{X}|Y}(\cdot|y)$, such that $\hat{X}$ adheres to the Markov chain $X \to Y \to \hat{X}$ (*i.e.* $X$ and $\hat{X}$ are statistically independent given $Y$).

### 2.1 DISTORTION AND PERCEPTUAL INDEX

Image restoration algorithms are typically evaluated by their average distortion $\mathbb{E}[\Delta(X, \hat{X})]$, where $\Delta(x, \hat{x})$ is some distortion measure that quantifies the discrepancy between $x$ and $\hat{x}$, and the expectation is taken over the joint distribution $p_{X,\hat{X}}$. Common examples for $\Delta(x, \hat{x})$ are the absolute error $\|x - \hat{x}\|_1$, the squared error $\|x - \hat{x}\|^2$, and LPIPS (Zhang et al., 2018). Moreover, as the goal in PIR is to produce reconstructions that would look natural to humans, PIR algorithms are also evaluated by perceptual quality measures. The ideal way to evaluate perceptual quality is to assess the ability of humans to distinguish between samples of ground-truth images and samples of reconstructed ones. This is typically done by conducting experiments where human observers vote on whether the generated images are real or fake (Dahl et al., 2017; Denton et al., 2015; Guadarrama et al., 2017; Iizuka et al., 2016; Isola et al., 2017; Salimans et al., 2016; Zhang et al., 2016; 2017). However, such experiments are too costly and impractical for optimizing models. A practical and sensible alternative to quantify the perceptual quality is via some *perceptual index* $d(p_X, p_{\hat{X}})$, where $d(\cdot, \cdot)$ is a statistical divergence between probability distributions (*e.g.*, Kullback–Leibler, Wasserstein) (Blau & Michaeli, 2018). Quantifying the perceptual index for high-dimensional distributions is both statistically and computationally intractable, so it is common to resort to approximations. Popular examples include the Fréchet Inception Distance (FID) (Heusel et al., 2017) and the Kernel Inception Distance (KID) (Bińkowski et al., 2018).

## 2.2 OPTIMAL ESTIMATORS FOR THE SQUARED ERROR DISTORTION

Due to the distortion-perception tradeoff (Blau & Michaeli, 2018), it has become common practice to compare image restoration algorithms on the distortion-perception plane, where the goal is to obtain *optimal* estimators with the lowest possible distortion given a prescribed level of perceptual index. This goal can be formalized by the distortion-perception function (Blau & Michaeli, 2018),

$$D(P) = \min_{p_{\hat{X}|Y}} \mathbb{E}[\Delta(X, \hat{X})] \quad \text{s.t.} \quad d(p_X, p_{\hat{X}}) \leq P. \tag{1}$$

Perhaps the most common points of interest on $D(P)$ are $D(\infty)$ and $D(0)$, where the first point corresponds to the estimator achieving minimal average distortion under no constraint, and the second corresponds to the estimator achieving minimal average distortion under a perfect perceptual index constraint. Considering the squared error distortion, these points are defined by

$$\min_{p_{\hat{X}|Y}} \mathbb{E}[\|X - \hat{X}\|^2] \quad \text{and} \tag{2}$$

$$\min_{p_{\hat{X}|Y}} \mathbb{E}[\|X - \hat{X}\|^2] \quad \text{s.t.} \quad p_{\hat{X}} = p_X, \tag{3}$$

respectively. It is well-known that the unique solution to Problem (2) is the posterior mean $\hat{X}^* := \mathbb{E}[X|Y]$, which typically produces overly-smooth reconstructions (Blau & Michaeli, 2018). Therefore, in PIR tasks, it is more appropriate to aim for the solution to Problem (3). Interestingly, Freirich et al. (2021) proved that a solution to Problem (3) can be obtained by solving the optimal transport problem

$$p_{U,V} \in \underset{p_{U',V'} \in \Pi(p_X, p_{\hat{X}^*})}{\arg\min} \mathbb{E}[\|U' - V'\|^2], \tag{4}$$

where $\Pi(p_X, p_{\hat{X}^*}) := \{p_{U',V'} : p_{U'} = p_X, p_{V'} = p_{\hat{X}^*}\}$ is the set of all joint probabilities $p_{U',V'}$ with marginals $p_{U'} = p_X$ and $p_{V'} = p_{\hat{X}^*}$. Namely, the optimal solution to Problem (3) can be constructed as follows: Given a degraded measurement $y$, first predict the posterior mean $\hat{x}^* = \mathbb{E}[X|Y = y]$, and then sample from $p_{U|V}(\cdot|\hat{x}^*)$, which is the optimal transport plan from $p_{\hat{X}^*}$ to $p_X$. Similarly to Freirich et al. (2021), we denote such a solution to Problem (3) by $\hat{X}_0$.

As discussed before, one of the most common and appealing solutions for PIR tasks is the estimator $\hat{X}$ that samples from the posterior distribution $p_{X|Y}$, such that $p_{\hat{X}|Y} = p_{X|Y}$. While such an estimator always attains a perfect perceptual index (Blau & Michaeli, 2018), its MSE is typically *larger* than that of $\hat{X}_0$ (Blau & Michaeli, 2018; Freirich et al., 2021) (see Figure 1). In other words, to design an algorithm with minimal MSE under a perfect perceptual index constraint, one should often *not* resort to posterior sampling, but rather to solving Problem (3). This is our goal in this paper. Lastly, one may wonder whether sampling from $p_{X|\hat{X}^*}$ instead of using the optimal transport plan from Equation (4) may also be effective in terms of MSE. However, in Appendix A.1 we prove that such an approach leads to precisely the same MSE as sampling from the posterior.

## 2.3 FLOW MATCHING AND RECTIFIED FLOWS

**Flow matching.** Flow matching algorithms (Albergo & Vanden-Eijnden, 2023; Lipman et al., 2023; Liu et al., 2023) are generative models defined via the ODE

$$dZ_t = v(Z_t, t)dt, \tag{5}$$

where $v$ is often called a *vector field*, and $Z_t$ is some forward process such that $p_{Z_0}$ is the source distribution, from which we can easily sample (*e.g.*, isotropic Gaussian noise), and $p_{Z_1}$ is the target distribution from which we aim to sample (*e.g.*, natural images). In principle, one can generate samples from the target distribution $p_{Z_1}$ by solving Equation (5), where samples from the source distribution $p_{Z_0}$ are set as the initial conditions for the ODE solver. Nevertheless, given a particular forward process $Z_t$, there are possibly many different vector fields that satisfy Equation (5). The goal in flow matching is to somehow find an appropriate vector field with desirable practical and theoretical properties, *e.g.*, where the solution to Equation (5) is unique.

**Rectified flow.** Rectified flow (Liu et al., 2023) is a flow matching algorithm defined via the particular forward process

$$Z_t = tZ_1 + (1 - t)Z_0, \tag{6}$$

---

**Algorithm 1:** Posterior-Mean Rectified Flow (PMRF)

---

**Training**

    *Stage 1:* Solve $\omega^* \leftarrow \arg\min_\omega \mathbb{E}\left[\|X - f_\omega(Y)\|^2\right]$

    *Stage 2:* Solve $\theta^* \leftarrow \arg\min_\theta \mathbb{E}\left[\|(X - Z_0) - v_\theta(Z_t, t)\|^2\right]$

    `// ` $Z_t \coloneqq tX + (1-t)(f_{\omega^*}(Y) + \sigma_s\epsilon)$`, where $t$ is sampled from $U[0,1]$.`

**Inference (using Euler's method with $K$ steps to solve the ODE)**

    Sample $\epsilon \sim \mathcal{N}(0, I)$

    $\hat{x} \leftarrow f_{\omega^*}(y) + \sigma_s\epsilon$                    `// $y$ is the given degraded measurement`

    **for** $i \leftarrow 0, \ldots, K-1$ **do**

        $\hat{x} \leftarrow \hat{x} + \frac{1}{K}v_{\theta^*}(\hat{x}, \frac{i}{K})$

    Return $\hat{x}$

---

which connects samples from $p_{Z_1}$ and $p_{Z_0}$ with straight lines. Here, $Z_0$ and $Z_1$ can be statistically independent, as is typically the case when learning a flow model from Gaussian noise to image data, but they can also have any joint distribution $p_{Z_0, Z_1}$. This forward process clearly adheres to the ODE $dZ_t = (Z_1 - Z_0)dt$, where $Z_1 - Z_0$ is the corresponding vector field. However, this is not a practical generative model, since it requires knowing the "destination" realization of $Z_1$ at any time step $t < 1$ (*i.e.*, the solution is not causal). To solve this issue, Liu et al. (2023) offer instead to use

$$v_{\text{RF}}(Z_t, t) = \mathbb{E}[Z_1 - Z_0 | Z_t], \tag{7}$$

which is a causal vector field that generates the target distribution, given that the solution to Equation (5) exists and is unique when adopting such a vector field (Theorem 3.3 in (Liu et al., 2023)). Interestingly, solving the ODE in Equation (5) with $v_{\text{RF}}$ often approximates the optimal transport map from the source distribution to the target one, especially when the process is repeated several times (*i.e.*, reflow) or when $p_{Z_1, Z_0}$ is *close* to the optimal transport plan between $p_{Z_0}$ and $p_{Z_1}$ (Liu et al., 2023; Tong et al., 2024). To learn $v_{\text{RF}}$, one can simply train a model $v_\theta$ by minimizing the loss

$$\int_0^1 \mathbb{E}\left[\|(Z_1 - Z_0) - v_\theta(Z_t, t)\|^2\right] dt, \tag{8}$$

where the expectation is taken over the joint distribution $p_{Z_1, Z_0}$ (Liu et al., 2023).

## 3   POSTERIOR-MEAN RECTIFIED FLOW

We now describe our proposed algorithm, which we coin Posterior-Mean Rectified Flow (PMRF) (Algorithm 1). Our method consists of two simple training stages. First, we train a model $f_\omega$ to predict the posterior mean by minimizing the MSE loss,

$$\omega^* = \arg\min_\omega \mathbb{E}\left[\|X - f_\omega(Y)\|^2\right]. \tag{9}$$

Note that this training stage can often be skipped, whenever there exists an off-the-shelf algorithm that attains sufficiently small MSE (high PSNR) in the desired restoration task. In the second stage, we train a rectified flow model $v_\theta$ (a vector field) to solve

$$\theta^* = \arg\min_\theta \int_0^1 \mathbb{E}\left[\|(X - Z_0) - v_\theta(Z_t, t)\|^2\right] dt, \tag{10}$$

where $Z_t \coloneqq tX + (1-t)Z_0$. Here, $Z_0 \coloneqq f_{\omega^*}(Y) + \sigma_s\epsilon$, where $\epsilon \sim \mathcal{N}(0, I)$ is statistically independent of $Y$ and $X$, and $\sigma_s$ is a hyper-parameter that controls the level of the Gaussian noise added to the posterior mean prediction. As shown by Albergo et al. (2023), adding such a noise is critical when the source and target distributions lie on low and high dimensional manifolds, respectively. Specifically, it alleviates the singularities resulting from learning a deterministic mapping between such distributions. Note, however, that adding noise to $f_{\omega^*}(Y)$ may harm the MSE of the reconstructions produced by PMRF, and so $\sigma_s$ should be taken to be sufficiently small.

To explain why PMRF approximates the desired estimator $\hat{X}_0$, we prove an important proposition and demonstrate it on a simple example with closed-form solutions. Specifically, let

$$d\hat{Z}_t = v_{\text{RF}}(\hat{Z}_t, t)dt \quad \text{with} \quad \hat{Z}_0 = Z_0 \tag{11}$$

be the ODE in PMRF, where $v_{\text{RF}}(z, t) = \mathbb{E}[X - Z_0|Z_t = z]$ and $\hat{Z}_t$ is the random vector generated by PMRF at time step $t \in [0, 1]$. In Appendix A.2 we prove the following:

**Proposition 1.** *Suppose that $\sigma_s = 0$, and let us assume that the solution of the ODE in Equation* (11) *exists and is unique. Then,*

(a) $\hat{Z}_1$ *attains a perfect perceptual index* ($p_{\hat{Z}_1} = p_X$).

(b) *The MSE of $\hat{Z}_1$ cannot be larger than that of the posterior sampler.*

(c) *If the distribution of $(X - \hat{X}^*)|Z_t = z_t$ is non-degenerate for almost every $z_t \in \text{supp}\, p_{Z_t}$ and $t \in [0, 1]$, then the MSE of $\hat{Z}_1$ is strictly smaller than that of the posterior sampler.*

Note that the assumption in *(a)* and *(b)* is the same as the one in (Liu et al., 2023), so it is not more limiting. Whether the additional assumption in *(c)* holds depends on the nature of the restoration task. For example, if $X$ can be restored from $Y$ with zero error (*i.e.*, $p_{X|Y}(\cdot|y)$ is a Dirac delta function for almost every $y$), then $X - \hat{X}^* = 0$ almost surely and the assumption in *(c)* does not hold. Yet, this is not an interesting setting as the degradation is not invertible in most practical scenarios. To gain intuition into a more common scenario, consider the following example:

**Example 1.** *Let $Y = X + N$, where $X \sim \mathcal{N}(0, 1)$ and $N \sim \mathcal{N}(0, \sigma_N^2)$ are statistically independent and $\sigma_N > 0$. Then, the MSE of $\hat{X}_0$ is strictly smaller than that of the posterior sampler. Moreover, when $\sigma_s = 0$, all the assumptions in Proposition 1 hold, and we have $\hat{Z}_1 = \hat{X}_0$ almost surely.*

See Appendix A.3 for the proof of Example 1. This example shows that PMRF not only outperforms posterior sampling, but may even *coincide* with the desired estimator $\hat{X}_0$ in certain cases.

## 4 RELATED WORK

Before moving on to demonstrate the effectiveness of our approach, it is instructive to note the difference between our PMRF method and existing techniques that may superficially seem similar.

**Diffusion and flow-based posterior samplers.** Diffusion or flow-based image restoration algorithms often attempt to sample from the posterior distribution by training a *conditional* model that takes $Y$ (or some function of $Y$, like $\hat{X}^*$) as an additional input (Lin et al., 2024; Zhu et al., 2024). Some works avoid training a conditional model for each task separately, and rather modify the sampling process of a trained unconditional diffusion model (Chung et al., 2023; Kawar et al., 2022). In Section 5.2 we perform a controlled experiment on various inverse problems, which shows that our PMRF method consistently outperforms posterior samplers with the same architecture.

**Flow from degraded image.** Some diffusion/flow models are trained on corresponding pairs of ground-truth images and degraded measurements (Albergo et al., 2023; Delbracio & Milanfar, 2023; Li et al., 2023). In this approach, the idea is to obtain a high-quality image by solving an ODE/SDE with the *degraded measurement* set as the initial condition. For example, Albergo et al. (2023) trained a rectified flow model for the forward process $Z_t = tX + (1 - t)Y^\dagger$, where $Y^\dagger$ is an upsampled version of $Y$ such that it matches the dimensionality of $X$. These algorithms are closely related to PMRF, in the sense that they learn to transport an *intermediate* signal (instead of pure noise) to the ground-truth image distribution. Yet, they have two critical disadvantages compared to PMRF. First, the flow model's design is not agnostic to the type of degradation, as the degraded signals can have varying dimensionalities or lie in a different domain than that of the ground-truth images (*e.g.*, in MRI image reconstruction). Thus, the task of the flow model may be harder than necessary, as it needs to *translate* signals from one domain to another. On the other hand, in PMRF the flow model always operates in the image domain, where the dimensionalities of the source and target signals are the same. Second, the *theoretical* motivation for flowing from $Y$ is not clear, at least from a reconstruction performance standpoint (*e.g.*, distortion). In contrast, the theoretical motivation underlying PMRF is clear: it approximates $\hat{X}_0$, which achieves the minimal possible MSE under the constraint of perfect perceptual index. As we show in Section 5.2, PMRF always either outperforms or is on-par with the solution that flows from $Y$ (see Figure 4).

**Methods that aim for $\hat{X}_0$ directly.** To the best of our knowledge, Deep Optimal Transport (DOT) (Adrai et al., 2023) is the only existing method that, like PMRF, attempts to approximate

$\hat{X}_0$ directly. Specifically, DOT approximates the desired optimal transport map (Equation (4)) via a linear transformation in the latent space of a variational auto-encoder (VAE) (Kingma & Welling, 2014). This transformation is computed in closed-form using the empirical means and covariances (in latent space) of the source distribution (that of the posterior mean predictions) and the target distribution (that of the ground-truth images), under the assumption that both are Gaussian. This method is computationally efficient, but the use of a VAE imposes a performance ceiling. Moreover, the optimal transport in DOT occurs in latent space and assumes that the source and target distributions are Gaussians, unlike Equation (4) which occurs in pixel space and does not make such an assumption. In contrast, PMRF does not use a VAE, and approximates the optimal transport directly in pixel space. In Section 5 we show that PMRF significantly outperforms DOT (see Figure 4).

## 5 EXPERIMENTS

### 5.1 BLIND FACE IMAGE RESTORATION

We train PMRF to solve the challenging blind face image restoration task, and compare its performance with leading methods. As in previous works (*e.g.*, (Wang et al., 2021)), we use the FFHQ data set (Karras et al., 2019) with images of size $512 \times 512$ to train our model. Similarly to previous works, we adopt a complex and random degradation process to synthesize the degraded images,

$$Y = [(X \circledast k_\sigma) \downarrow_R + N_\delta]_{\text{JPEG}_Q}, \tag{12}$$

where $\circledast$ denotes convolution, $k_\sigma$ is a Gaussian blur kernel of size $41 \times 41$ and variance $\sigma^2$, $\downarrow_R$ is bilinear down-sampling by a factor $R$, $N_\delta$ is white Gaussian noise of variance $\delta^2$, and $[\cdot]_{\text{JPEG}_Q}$ is JPEG compression-decompression with quality factor $Q$. Similarly to (Yue & Loy, 2024), we synthesize the degraded images by sampling $\sigma, R, \delta$ and $Q$ uniformly from $[0.1, 15]$, $[0.8, 32]$, $[0, 20]$, and $[30, 100]$, respectively. See Appendix B.1 for additional implementation details.

### 5.1.1 EVALUATION SETTINGS

For evaluation, we consider the common synthetic CelebA-Test benchmark, as well as the real-world data sets LFW-Test (Huang et al., 2008; Wang et al., 2021), WebPhoto-Test (Wang et al., 2021), CelebAdult-Test (Wang et al., 2021), and WIDER-Test (Zhou et al., 2022). CelebA-Test consists of 3,000 high-quality images taken from the test partition of CelebA-HQ (Karras et al., 2018), and the degraded images were synthesized by Wang et al. (2021). For the real-world data sets, the degradations are unknown and there is no access to the clean ground-truth images. We compare our performance with DOT (Adrai et al., 2023) and leading blind face restoration models, including BFRffussion (Chen et al., 2024), DiffBIR (Lin et al., 2024), DifFace (Yue & Loy, 2024), CodeFormer (Zhou et al., 2022), GFPGAN (Wang et al., 2021), VQFRv1 and VQFRv2 (Gu et al., 2022), RestoreFormer and RestoreFormer++ (Wang et al., 2022; 2023b). We do not compare with FlowIE (Zhu et al., 2024), as the official checkpoints of this method are currently unavailable. However, note that FlowIE is a *conditional* method that employs a ControlNet (similarly to DiffBIR). Namely, it falls under the category of methods that attempt to sample from the posterior distribution, which are fundamentally different from PMRF. Notably, the restoration methods that we compare against also use the degradation model from Equation (12), though the ranges of $\sigma$, $R$, $\delta$, and $Q$ differ across methods. The ranges we choose, those from (Yue & Loy, 2024), are the most *severe* among all the compared methods. For example, the range of $R$ we use is $[0.8, 32]$, whereas Wang et al. (2021) use $[1, 8]$. Thus, PMRF attempts to solve a more difficult restoration task than some of the compared methods. In the following experiments, we use $K = 25$ flow steps in PMRF (Algorithm 1). Refer to Appendix B.2 for an evaluation of additional values of $K$, and to Appendix B.3 for the implementation details of DOT.

### 5.1.2 RESULTS ON CELEBA-TEST

For the CelebA-Test benchmark, we measure the perceptual quality by FID (Heusel et al., 2017), KID (Bińkowski et al., 2018), NIQE (Mittal et al., 2013), and Precision (Kynkäänniemi et al., 2019), and measure the distortion by the PSNR, SSIM (Wang et al., 2004), and LPIPS (Zhang et al., 2018). Similarly to previous works (Gu et al., 2022; Wang et al., 2021), we also compute the identity metric Deg (using the embedding angle of ArcFace (Deng et al., 2019)) and the landmark distance LMD.

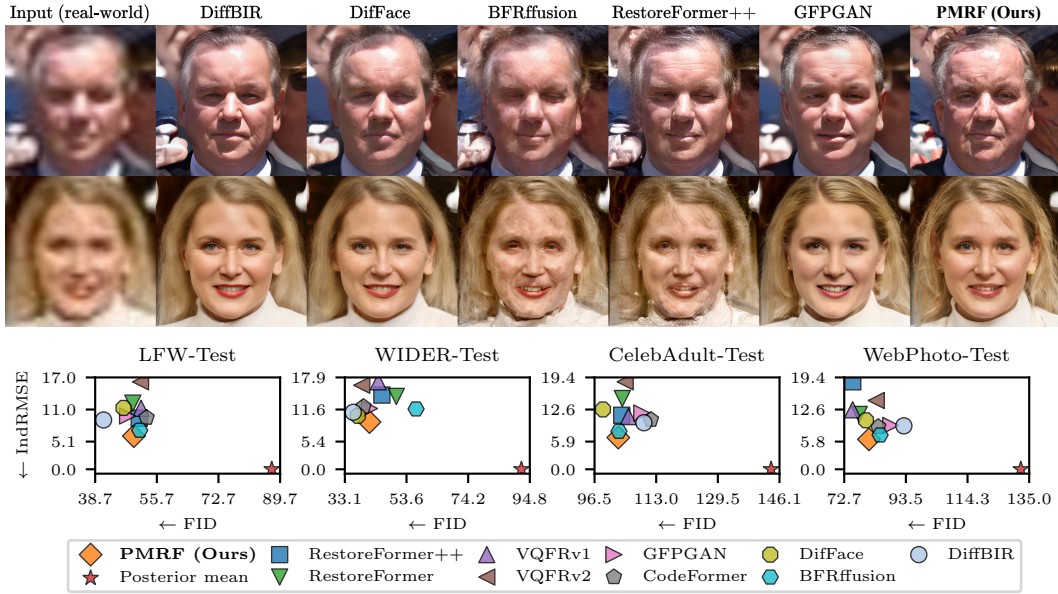

Figure 3: Real-world face image restoration. **Top**: Qualitative results on inputs from the **WIDER-Test** data set. **Bottom**: Comparison on the "distortion"-perception plane (IndRMSE *vs.* FID), where IndRMSE *indicates* the RMSE of each method (the true distortion cannot be computed as there is no access to the ground-truth images). Our algorithm outperforms all other methods in IndRMSE, while achieving on-par perceptual quality compared to the state-of-the-art.

Both of these can be considered as distortion measures, as they quantify some type of discrepancy between each reconstructed image and its ground-truth counterpart.

The results are reported in Table 1. Notably, PMRF outperforms all other methods in FID, KID, PSNR, and SSIM, achieves the second best scores in NIQE, Precision and Deg, and the third best scores in LPIPS, and LMD. Interestingly, no other method attains such a *consensus* in performance like PMRF, namely, where none of the measures are significantly compromised compared to the state-of-the-art. For example, while DifFace achieves the highest Precision, it attains worse LMD, Deg, LPIPS, SSIM, and PSNR compared to the third best method in each of these metrics. This demonstrates that PMRF produces robust reconstructions, in the sense that it does not "over-fit" particular perceptual quality or distortion measures, but rather achieves high performance in all of them simultaneously. Visual results are provided in Figure 2 and in Figure 6 in the appendix.

### 5.1.3 RESULTS ON REAL-WORLD DEGRADED IMAGES

Evaluating the distortion for real-world degraded images is impossible, as there is no access to the ground-truth images. Consequently, previous works conduct only a perceptual quality evaluation (*e.g.*, FID) on real-world data sets such as WIDER-Test and LFW-Test. Yet, high perceptual quality alone is clearly not indicative of reconstruction performance (to attain high perceptual quality, one may simply ignore the inputs and generate samples from $p_X$). Thus, we consider a measure which *indicates* the Root MSE (RMSE) and allows ranking algorithms according to their (approximate) RMSE, *without* access to the ground-truth images. Specifically, for any estimator $\hat{X}$ it holds that

$$\mathbb{E}[\|X - \hat{X}\|^2] \approx \mathbb{E}[\|\hat{X} - f(Y)\|^2] + m, \tag{13}$$

where $f(Y) \approx \hat{X}^*$ is an approximation of the true posterior mean predictor $\hat{X}^*$, and $m$ is a constant that does not depend on $\hat{X}$ (see Appendix E for an explanation). Thus, the square root of $\mathbb{E}[\|\hat{X} - f(Y)\|^2]$, which we denote by IndRMSE, *indicates* the true RMSE. We utilize the posterior mean predictor trained by (Yue & Loy, 2024)[2] as $f$, and compute the IndRMSE of all the evaluated algorithms on the LFW-Test, WebPhoto-Test, CelebAdult-Test, and WIDER-Test data sets. As

---

[2]Importantly, the *exact* same posterior mean predictor model (and weights) is also used by other methods such as DifFace and DiffBIR, so this is a fair evaluation.

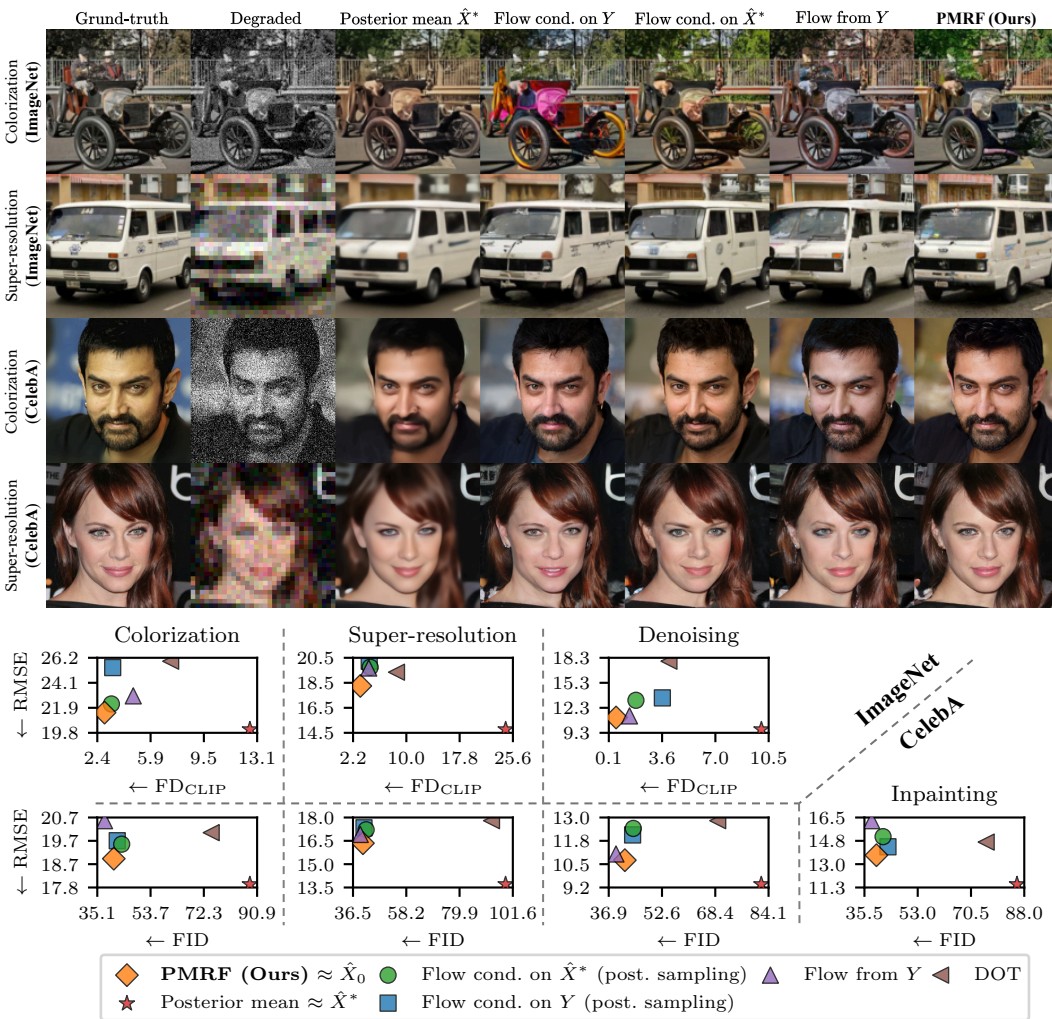

Figure 4: A controlled experiment comparing PMRF (our method) with several baseline methods, where the models are trained with the same architecture, hyper-parameters, *etc.* (see Section 5.2). **Top**: Qualitative comparison of PMRF and the baseline methods on several tasks. **Bottom**: Quantitative comparison on the distortion-perception plane. DOT is not a flow model, but rather another approach that attempts to approximate $\hat{X}_0$ (like PMRF). These experiments demonstrate that PMRF is either superior or is on-par with previous frameworks (*i.e.*, posterior sampling or flowing from $Y$) on a variety of image restoration tasks. See Section 5.2 for more details.

before, we evaluate perceptual quality by FID, KID, NIQE, and Precision. In Figure 3 we provide visual results on inputs from the WIDER-Test data set, and compare the algorithms on a "distortion"-perception plane (IndRMSE *vs.* FID). DOT is not plotted as it achieves far worse FID compared to other methods. Our algorithm attains the best (smallest) IndRMSE on all data sets, while achieving on-par perceptual quality compared to the state-of-the-art. This indicates that PMRF achieves superior distortion on such real-world data sets, while not compromising perceptual quality. In the appendix, we report the rest of the perceptual quality measures in Tables 7 to 10, provide visual results in Figures 7 to 9, and also report the performance of DOT.

## 5.2 COMPARING PMRF WITH PREVIOUS FRAMEWORKS IN CONTROLLED EXPERIMENTS

One may wonder whether the performance of PMRF is attributed to the framework *itself* (Algorithm 1), or, maybe it is attributed to the model architecture, the rectified flow training approach, the chosen hyper-parameters, *etc.* Could we have done better by training a flow to sample from the

posterior, or by adopting the approach of (Albergo et al., 2023) and flow directly from $Y$? Here, we conduct a controlled study where we demonstrate that the high performance of PMRF is indeed attributed to the proposed framework itself (Algorithm 1). Specifically, we consider several image restoration tasks (denoising, super-resolution, *etc.*), where we train PMRF and several baseline methods on the "same grounds", using the ImageNet (Deng et al., 2009) ($128 \times 128$) and FFHQ ($256 \times 256$) data sets. In each task we train two *conditional* rectified flow models, where one is conditioned on the degraded measurement $Y$ (we call this method *flow conditioned on $Y$*), and the other is conditioned on the posterior mean predictor $f_{\omega^*}(Y)$ (we call this method *flow conditioned on $\hat{X}^*$*). The first model represents posterior sampling methods, and the second model allows for a fair comparison of model capacity with PMRF (since PMRF is comprised of $f_{\omega^*}(Y)$ and a flow model). In fact, theoretically speaking, the second approach achieves precisely the same MSE as the posterior sampler (see Appendix A.1), and is often used in practice (*e.g.*, in (Lin et al., 2024; Zhu et al., 2024)). In addition, we train an *unconditional* rectified flow model, where the forward process is defined as $Z_t = tX + (1-t)Z_0$, $Z_0 = Y^\dagger + \sigma_s\epsilon$, $\epsilon \sim \mathcal{N}(0, I)$, and $Y^\dagger$ is the up-scaled version of the degraded measurement $Y$ such that it matches the dimensionality of $X$ (we call this method *flow from $Y$*). This method represents the frameworks in (Albergo et al., 2023; Delbracio & Milanfar, 2023; Li et al., 2023), which we discuss in Section 4. All of the models are trained with the same hyper-parameters as PMRF, using the same architecture, learning rate, weight decay, number of training epochs, *etc*. Moreover, for PMRF and flow conditioned on $\hat{X}^*$ method, we use the exact same architecture and weights for $f_{\omega^*}(Y)$. To clarify the differences between the mathematical formulations of the baseline methods, in Table 11 in the appendix we summarize the definitions of the training loss and the forward process of all methods. Moreover, in Algorithms 2 to 4 we disclose a pseudo-code for the training and inference procedures of the baseline methods. While DOT is not a flow method, we still evaluate its performance as it is related to PMRF.

In Figure 4 we compare the algorithms on the distortion-perception plane (RMSE *vs.* FID for face restoration, and RMSE *vs.* FD$_{\text{CLIP}}$ (Stein et al., 2023) for ImageNet restoration), using $K = 100$ flow steps for each flow algorithm. We clearly PMRF *dominates* all other methods in most tasks, achieving notably smaller RMSE without compromising (and sometimes even *improving*) perceptual quality. This demonstrates that PMRF achieves our desired goal, which is to attain low distortion without compromising on perceptual quality. For the image denoising tasks, we observe that PMRF and flow from $Y$ attain relatively similar performance, and both dominate the posterior sampling approaches. We hypothesize that, in some tasks (*e.g.*, denoising), flowing from $Y$ may be as effective as PMRF in terms of approximating $\hat{X}_0$. To demonstrate this, we prove in Appendix D that flowing from $Y$ is optimal in the toy problem in Example 1 (just like PMRF). Yet, our experiments demonstrate that PMRF generally leads to better performance compared to previous frameworks. To assess the effectiveness of each method given different inference time constraints, in Figure 5 in the appendix we vary the number of flow inference steps $K$ for each method. Interestingly, we observe that PMRF is still either on-par or dominates the other methods for *any* given number of inference steps. These results further demonstrate that the superior performance of PMRF is attributed to our framework itself, rather than to the chosen hyper-parameters. See Appendix C for more details, and refer to Figures 10 to 16 in the appendix for visual comparisons.

## 6 CONCLUSION AND LIMITATIONS

We presented a method that directly approximates $\hat{X}_0$ – the estimator that minimizes the MSE under a perfect perceptual index constraint (Equation (3)). We showed that our approach, coined PMRF, is a simple yet highly effective image restoration algorithm that outperforms previous frameworks (*e.g.*, posterior sampling, flow from $Y$, and GAN-based methods) in a variety of image restoration tasks. As we explained in Section 3, PMRF alleviates the issues resulting from solving the ODE by adding Gaussian noise to the posterior mean predictions. We note that the noise level $\sigma_s$ should be carefully tuned, as taking it to be too large or too small may cause the MSE or the perceptual quality of PMRF to degrade, respectively. While the flow from $Y$ method (Algorithm 4) suffers from the same limitation (though it does not provide a theoretical guarantee on the MSE, like PMRF), this may be considered a disadvantage of PMRF compared to posterior sampling methods (*e.g.*, Algorithm 2), which do not require such a hyper-parameter. Finally, we proved in Proposition 1 that, under some conditions, PMRF is guaranteed to achieve a smaller MSE than the posterior sampler. However, as in (Liu et al., 2023), one could argue that the assumptions in Proposition 1 may be too limiting in some cases.

## REPRODUCIBILITY STATEMENT

Our codes are available at `https://github.com/ohayonguy/PMRF`. We provide all the explanations and checkpoints necessary to reproduce our results, including training, inference, and the computation of the distortion and perceptual quality measures in Section 5. Besides our code, our paper discloses all the implementation details required to reproduce the results, including architecture details, training hyper-parameters, *etc*. Refer to Sections 5.1 and 5.2 and appendices B and C for implementation details, and to Table 12 in the appendix for a summary of our training hyper-parameters.

## ACKNOWLEDGMENTS

This research was partially supported by the Israel Science Foundation (ISF) under Grants 2318/22, 951/24 and 409/24, and by the Council for Higher Education – Planning and Budgeting Committee.

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

# A    SUPPLEMENTARY EXPLANATIONS FOR PMRF

## A.1    PROOF THAT CONDITIONING ON $\hat{X}^*$ ACHIEVES THE SAME MSE AS POSTERIOR SAMPLING

**Proposition 2.** *Let $\hat{X}'$ be the estimator which, given any degraded measurement $y$, first predicts the posterior mean $\hat{x}^* = \mathbb{E}[X|Y = y]$ and then samples from $p_{X|\hat{X}^*}(\cdot|\hat{x}^*)$[3]. Then, the MSE of $\hat{X}'$ equals twice the MMSE, which is the MSE attained by the posterior sampler.*

*Proof.* The MSE of $\hat{X}'$ is given by

$$\mathbb{E}[\|X - \hat{X}'\|^2] = \mathbb{E}[\|X - \hat{X}^*\|^2] + \mathbb{E}[\|\hat{X}' - \hat{X}^*\|^2], \tag{14}$$

where this equality follows from Lemma 2 in (Freirich et al., 2021) (Appendix B.1). By the definition of $\hat{X}'$ we have $p_{\hat{X}', \hat{X}^*} = p_{X, \hat{X}^*}$, so

$$\mathbb{E}[\|\hat{X}' - \hat{X}^*\|^2] = \mathbb{E}[\|X - \hat{X}^*\|^2]. \tag{15}$$

Substituting this result into Equation (14), we get

$$\mathbb{E}[\|X - \hat{X}'\|^2] = 2\mathbb{E}[\|X - \hat{X}^*\|^2]. \tag{16}$$

Namely, $\hat{X}'$ attains precisely the same MSE as the posterior sampler, which is equal to twice the MMSE (Blau & Michaeli, 2018). Thus, in theory, one should not expect to improve the MSE of a conditional diffusion/flow model by supplying $\hat{X}^*$ as a condition instead of $Y$. $\square$

## A.2    PROOF OF PROPOSITION 1

For completeness, we first restate Proposition 1 and then provide its proof.

**Proposition 1.** *Suppose that $\sigma_s = 0$, and let us assume that the solution of the ODE in Equation (11) exists and is unique. Then,*

*(a) $\hat{Z}_1$ attains a perfect perceptual index ($p_{\hat{Z}_1} = p_X$).*
*(b) The MSE of $\hat{Z}_1$ cannot be larger than that of the posterior sampler.*
*(c) If the distribution of $(X - \hat{X}^*)|Z_t = z_t$ is non-degenerate for almost every $z_t \in \operatorname{supp} p_{Z_t}$ and $t \in [0, 1]$, then the MSE of $\hat{Z}_1$ is strictly smaller than that of the posterior sampler.*

*Proof.* We first prove *(a)* and *(b)* assuming that the solution for the ODE in Equation (11) exists and is unique for $\sigma_s = 0$. Then, we will prove *(c)* by also assuming that the distribution of $(X - \hat{X}^*)|Z_t = z_t$ is non-degenerate for almost every $z_t$ and $t \in [0, 1]$.

From Theorem 3.3 in (Liu et al., 2023) we have $p_{\hat{Z}_t} = p_{Z_t}$ for every $t \in [0, 1]$. This implies that $p_{\hat{Z}_1} = p_{Z_1} = p_X$, *i.e.*, PMRF attains a perfect perceptual index when $\sigma_s = 0$. This proves *(a)*.

Next, without additional assumptions, we will prove *(b)* by showing that

$$\mathbb{E}[\|\hat{Z}_1 - \hat{X}^*\|^2] \leq \mathbb{E}[\|X - \hat{X}^*\|^2], \tag{17}$$

which will imply that the MSE of $\hat{Z}_1$ can only be smaller than that of the posterior sampler. Since $\sigma_s = 0$, we have $Z_0 = \hat{X}^* + \sigma_s \epsilon = \hat{X}^*$. Following similar arguments to those in the proof of

---

[3]Note that $\hat{X}'$ is a "posterior sampler" which is conditioned on $\hat{X}^*$. Thus, Algorithm 3 represents such an algorithm, which is one of the baseline methods we evaluate in Section 5.2.

Theorem 3.5 in (Liu et al., 2023), it holds that

$$\mathbb{E}[\|\hat{Z}_1 - \hat{X}^*\|^2] = \mathbb{E}\left[\left\|\int_0^1 v_{\mathrm{RF}}(\hat{Z}_t, t)dt\right\|^2\right] \tag{18}$$

$$= \mathbb{E}\left[\left\|\int_0^1 v_{\mathrm{RF}}(Z_t, t)dt\right\|^2\right] \tag{19}$$

$$\leq \mathbb{E}\left[\int_0^1 \|v_{\mathrm{RF}}(Z_t, t)\|^2 dt\right] \tag{20}$$

$$= \mathbb{E}\left[\int_0^1 \left\|\mathbb{E}[X - \hat{X}^* | Z_t]\right\|^2 dt\right] \tag{21}$$

$$\leq \mathbb{E}\left[\int_0^1 \mathbb{E}[\|X - \hat{X}^*\|^2 | Z_t]dt\right] \tag{22}$$

$$= \int_0^1 \mathbb{E}\left[\mathbb{E}[\|X - \hat{X}^*\|^2 | Z_t]\right] dt \tag{23}$$

$$= \int_0^1 \mathbb{E}[\|X - \hat{X}^*\|^2]dt \tag{24}$$

$$= \mathbb{E}[\|X - \hat{X}^*\|^2], \tag{25}$$

where Equation (18) follows from the definition of $\hat{Z}_1$ and $\hat{X}^*$, Equation (19) follows from the fact that $p_{\hat{Z}_t} = p_{Z_t}$, Equation (20) follows from Jensen's inequality, Equation (21) follows from the definition of $v_{\mathrm{RF}}(Z_t, t)$, Equation (22) follows from Jensen's inequality, Equation (23) follows from the linearity of the integral operator, and Equation (24) follows from the law of total expectation. Thus, we have $\mathbb{E}[\|\hat{Z}_1 - \hat{X}^*\|^2] \leq \mathbb{E}[\|X - \hat{X}^*\|^2]$. Combining this result with Lemma 2 from (Freirich et al., 2021) (Appendix B.1), we conclude that

$$\mathbb{E}[\|X - \hat{Z}_1\|^2] = \mathbb{E}[\|X - \hat{X}^*\|^2] + \mathbb{E}[\|\hat{Z}_1 - \hat{X}^*\|^2]$$
$$\leq 2\mathbb{E}[\|X - \hat{X}^*\|^2], \tag{26}$$

where the left hand side is the MSE of PMRF, and the right hand side is the MSE of the posterior sampler, which always equals twice the MMSE (Blau & Michaeli, 2018).

Finally, to prove *(c)*, let us further assume that $(X - \hat{X}^*)|Z_t = z_t$ is a non-degenerate random vector for every $z_t \in \mathrm{supp}\, p_{Z_t}$ and $t \in [0, 1]$. Thus, the inequality in Equation (22) becomes strict (from Jensen's inequality for strictly convex functions), and hence we have $\mathbb{E}[\|\hat{Z}_1 - \hat{X}^*\|^2] < \mathbb{E}[\|X - \hat{X}^*\|^2]$. Combining this result with Lemma 2 from (Freirich et al., 2021) (Appendix B.1), we conclude that

$$\mathbb{E}[\|X - \hat{Z}_1\|^2] < 2\mathbb{E}[\|X - \hat{X}^*\|^2]. \tag{27}$$

Namely, the MSE of $\hat{Z}_1$ (left hand side) is strictly smaller than that of the posterior sampler (right hand side). □

### A.3 PROOF OF THE RESULTS IN EXAMPLE 1

From (Blau & Michaeli, 2018; Freirich et al., 2021), we know that $\hat{X}_0$ in Example 1 attains a MSE that is *strictly* smaller than that of the posterior sampler (assuming that $\sigma_N > 0$). Specifically, the closed-form solution of $\hat{X}_0$ in Example 1 is given by (Freirich et al., 2021):

$$\hat{X}_0 = \frac{1}{\sqrt{1 + \sigma_N^2}}Y. \tag{28}$$

Moreover, in this example, it is well known that the posterior mean $\mathbb{E}[X|Y]$ is given by

$$\hat{X}^* = \frac{1}{1 + \sigma_N^2}Y. \tag{29}$$

Next, we will prove that:

*(a)* All the assumptions in Proposition 1 hold.
*(b)* $\hat{Z}_1 = \hat{X}_0$ almost surely.

**Proof of (a).** Since $\sigma_s = 0$, we have $v_{\text{RF}}(Z_t, t) = \mathbb{E}[X - \hat{X}^*|Z_t]$ and $Z_t = tX + (1 - t)\hat{X}^*$. Below, we show that

$$\text{Cov}(X - \hat{X}^*, Z_t) = t\frac{\sigma_N^2}{1 + \sigma_N^2}, \quad \text{and} \tag{30}$$

$$\text{Var}(Z_t) = t^2\frac{\sigma_N^2}{1 + \sigma_N^2} + \frac{1}{1 + \sigma_N^2}. \tag{31}$$

Since $X - \hat{X}^*$ and $Z_t$ are jointly Gaussian[4], we have

$$\begin{aligned}
v_{\text{RF}}(Z_t, t) &= \mathbb{E}[X - \hat{X}^*|Z_t] \\
&= \mathbb{E}[X - \hat{X}^*] + \frac{\text{Cov}(X - \hat{X}^*, Z_t)}{\text{Var}(Z_t)}(Z_t - \mathbb{E}[Z_t]) \\
&= \frac{\text{Cov}(X - \hat{X}^*, Z_t)}{\text{Var}(Z_t)}Z_t, \tag{32} \\
&= \frac{t\frac{\sigma_N^2}{1+\sigma_N^2}}{t^2\frac{\sigma_N^2}{1+\sigma_N^2} + \frac{1}{1+\sigma_N^2}}Z_t \\
&= \frac{t\sigma_N^2}{1 + t^2\sigma_N^2}Z_t, \tag{33}
\end{aligned}$$

where Equation (32) follows from the fact that $\mathbb{E}[X - \hat{X}^*] = 0$ and $\mathbb{E}[Z_t] = 0$. One can verify that the solution of $d\hat{Z}_t = v_{\text{RF}}(\hat{Z}_t, t)dt$ for any initial condition $\hat{Z}_0 = c$ is unique and is given by

$$\hat{Z}_t = c\sqrt{1 + t^2\sigma_N^2}. \tag{34}$$

To show that the distribution of $(X - \hat{X}^*)|Z_t = z_t$ is non-degenerate for almost every $z_t$ and $t \in [0, 1]$, note that

$$\begin{aligned}
\text{Var}(X - \hat{X}^*) &= \text{Cov}(X - \hat{X}^*, X - \hat{X}^*) \\
&= \text{Cov}(X, X) - 2\text{Cov}(X, \hat{X}^*) + \text{Cov}(\hat{X}^*, \hat{X}^*) \\
&= 1 - 2\text{Cov}\left(X, \frac{1}{1 + \sigma_N^2}Y\right) + \text{Cov}\left(\frac{1}{1 + \sigma_N^2}Y, \frac{1}{1 + \sigma_N^2}Y\right) \\
&= 1 - \frac{2}{1 + \sigma_N^2}\text{Cov}(X, Y) + \frac{1}{(1 + \sigma_N^2)^2}\text{Cov}(Y, Y) \\
&= 1 - \frac{2}{1 + \sigma_N^2} + \frac{1}{1 + \sigma_N^2} \\
&= 1 - \frac{1}{1 + \sigma_N^2} \\
&= \frac{\sigma_N^2}{1 + \sigma_N^2}. \tag{35}
\end{aligned}$$

---

[4] $X - \hat{X}^*$ and $Z_t$ can be written as a linear transformation of $(X, Y)$, which are jointly Gaussian random variables. Thus, $X - \hat{X}^*$ and $Z_t$ are jointly Gaussian.

Thus, for any $t > 0$, and assuming $\sigma_N > 0$, the correlation between $X - \hat{X}^*$ and $Z_t$ is given by

$$
\begin{aligned}
\frac{\text{Cov}(X - \hat{X}^*, Z_t)}{\sqrt{\text{Var}(Z_t)\text{Var}(X - \hat{X}^*)}} &= \frac{t\frac{\sigma_N^2}{1+\sigma_N^2}}{\sqrt{\left(t^2 \frac{\sigma_N^2}{1+\sigma_N^2} + \frac{1}{1+\sigma_N^2}\right)\left(\frac{\sigma_N^2}{1+\sigma_N^2}\right)}} \\
&= \frac{t\sigma_N}{\sqrt{1 + t^2\sigma_N^2}} \\
&= \frac{1}{\sqrt{1 + \frac{1}{t^2\sigma_N^2}}} \\
&< 1.
\end{aligned}
\tag{36}
$$

Namely, the correlation between $X - \hat{X}^*$ and $Z_t$ is strictly smaller than 1 for every $t \in (0, 1]$. Moreover, for $t = 0$ the correlation between $X - \hat{X}^*$ and $Z_t$ clearly equals zero, so such a correlation is smaller than 1 for every $t \in [0, 1]$. This implies that the distribution of $(X - \hat{X}^*)|Z_t = z_t$ is non-degenerate for almost every $z_t$ and $t \in [0, 1]$, and so all the assumptions in Proposition 1 hold.

To prove Equations (30) and (31), first note that $\text{Cov}(X, \hat{X}^*) = \text{Cov}(\hat{X}^*, \hat{X}^*) = \frac{1}{1+\sigma_N^2}$, and so $\text{Cov}(X, \hat{X}^*) - \text{Cov}(\hat{X}^*, \hat{X}^*) = 0$. Thus,

$$
\begin{aligned}
\text{Cov}(X - \hat{X}^*, Z_t) &= \text{Cov}(X - \hat{X}^*, tX + (1 - t)\hat{X}^*) \\
&= t(\text{Cov}(X, X) - \text{Cov}(X, \hat{X}^*)) + (1 - t)(\text{Cov}(X, \hat{X}^*) - \text{Cov}(\hat{X}^*, \hat{X}^*)) \\
&= t\left(1 - \frac{1}{1 + \sigma_N^2}\right) \\
&= t\frac{\sigma_N^2}{1 + \sigma_N^2},
\end{aligned}
\tag{37}
$$

and,

$$
\begin{aligned}
\text{Var}(Z_t) &= \text{Cov}(Z_t, Z_t) \\
&= \text{Cov}(tX + (1 - t)\hat{X}^*, tX + (1 - t)\hat{X}^*) \\
&= t^2\text{Cov}(X, X) + 2t(1 - t)\text{Cov}(X, \hat{X}^*) + (1 - t)^2\text{Cov}(\hat{X}^*, \hat{X}^*) \\
&= t^2 + (2t(1 - t) + (1 - t)^2)\frac{1}{1 + \sigma_N^2} \\
&= t^2 + (2t - 2t^2 + 1 - 2t + t^2)\frac{1}{1 + \sigma_N^2} \\
&= t^2 + (1 - t^2)\frac{1}{1 + \sigma_N^2} \\
&= t^2\frac{\sigma_N^2}{1 + \sigma_N^2} + \frac{1}{1 + \sigma_N^2}.
\end{aligned}
\tag{38}
$$

**Proof of (b).** The proof follows directly from Equation (34). Specifically, for the initial condition $\hat{Z}_0 = \hat{X}^*$, we have

$$
\begin{aligned}
\hat{Z}_1 &= \sqrt{1 + \sigma_N^2}\hat{X}^* \\
&= \sqrt{1 + \sigma_N^2}\frac{1}{1 + \sigma_N^2}Y \\
&= \frac{1}{\sqrt{1 + \sigma_N^2}}Y \\
&= \hat{X}_0.
\end{aligned}
\tag{39}
\tag{40}
$$

Thus, in Example 1, PMRF with $\sigma_s = 0$ coincides with the desired optimal estimator $\hat{X}_0$.

### A.4 REFLOW (OPTIONAL)

To potentially improve the MSE of PMRF further, one may conduct a *reflow* procedure (Liu et al., 2023), where a sequence of flow models are trained, and the flow model at index $k+1$ learns to flow from the source distribution to the distribution generated by the flow model at index $k$. Specifically, let $\hat{Z}_1^{k+1}$ be the random vector generated by PMRF (Algorithm 1), where $\hat{Z}_1^k$ replaces the role of $X$ in Algorithm 1 and $\hat{Z}_1^0 = X$ ($Z_0$ remains unchanged). Thus, from Theorem 3.5 in (Liu et al., 2023), we have $\mathbb{E}[c(\hat{Z}_1^{k+1} - Z_0)] \leq \mathbb{E}[c(\hat{Z}_1^k - Z_0)]$, which implies the reflowing may only improve the MSE of PMRF, and hence improve the approximation of the desired optimal transport map (Equation 4). We leave this possibility for future work.

## B SUPPLEMENTARY DETAILS AND EXPERIMENTS IN BLIND FACE IMAGE RESTORATION

### B.1 IMPLEMENTATION DETAILS OF PMRF

During training, we only use random horizontal flips for data augmentation. We use the SwinIR (Liang et al., 2021) model trained by Yue & Loy (2024) as the posterior mean predictor $f_{\omega^*}$ in Algorithm 1, and use $\sigma_s = 0.1$. This model was trained using the same synthetic degradation as in Equation (12), with the same ranges for $\sigma$, $R$, $\delta$, and $Q$ we mentioned in Section 5.1. The SwinIR model's weights are kept frozen during the vector field's training stage, and the same weights are utilized during inference as well. The vector field $v_\theta$ is a HDiT model (Crowson et al., 2024), which we train from scratch. As in (Crowson et al., 2024), we sample $t$ uniformly from $U[0, 1]$ using a stratified sampling strategy. The vector field is trained for 3850 epochs using the AdamW optimizer (Loshchilov & Hutter, 2019), with a learning rate of $5 \cdot 10^{-4}$, $(\beta_1, \beta_2) = (0.9, 0.95)$, and a weight decay of $10^{-2}$ (as in (Crowson et al., 2024)). In the last 350 epochs, we reduce the learning rate gradually, multiplying it by $0.98$ at the end of every epoch. The training batch size is set to 256 and is kept fixed. We compute the exponential moving average (EMA) of the model's weights, using a decay of $0.9999$. The EMA weights of the model are then used in all evaluations. Our model is trained using bfloat16 mixed precision. A summary of the vector field training hyper-parameters is provided in Table 12.

### B.2 VARYING THE NUMBER OF FLOW STEPS $K$ IN PMRF

In Tables 2 to 6 we evaluate the performance of PMRF for various choices of $K$ (the number of inference steps in Algorithm 1). As expected, increasing $K$ generally improves the perceptual quality while harming the distortion.

### B.3 DETAILS OF DOT

We use the official codes of DOT (Adrai et al., 2023) as provided by the authors. This method performs optimal transport between the source and target distributions in latent space, using the closed-form solution for the optimal transport map between two Gaussians. As in (Adrai et al., 2023), we use the VAE (Kingma & Welling, 2014) of stable-diffusion (Rombach et al., 2022). For computing the latent empirical mean and covariance of the target distribution, we provide to the code the first 1000 images from FFHQ, with images of size $512 \times 512$ (the default is 100 images, so using 1000 images instead ensures that the performance of DOT is not compromised, as explain by Adrai et al. (2023)). For computing the latent empirical mean and covariance of the source distribution, we randomly synthesize degraded images according to Equation (12) from the first 1000 images in FFHQ, and reconstruct each image using the SwinIR model with the pre-trained weights from (Yue & Loy, 2024) (the same weights we use in PMRF). Given a degraded image $y$ at test time, the code of Adrai et al. (2023) first predicts the posterior mean using the SwinIR model, encodes it to latent space, optimally transports the result using the pre-computed empirical means and covariances, and finally uses the decoder to obtain the reconstructed image.

### B.4 Computation of FID, KID, and Precision

For each data set and algorithm, the FID, KID, and Precision are computed between the entire FFHQ $512 \times 512$ training set, and the reconstructed images produced for the degraded images in the *test* data set (as in previous works). For example, for the evaluations on the CelebA-Test data, this means that the FID is computed between the 70,000 FFHQ images, and the 3,000 CelebA-Test reconstructed images.

## C Supplementary details on Section 5.2

### C.1 Degradations

**Face restoration.** The degraded images in each face restoration task in the controlled experiments are synthesized according to the following degradations:

1. **Denoising**: We apply additive white Gaussian noise with standard deviation $0.35$.
2. **Super-resolution**: We use the $8\times$ bicubic down-sampling operator, and add Gaussian noise with standard deviation $0.05$.
3. **Inpainting**: We randomly mask $90\%$ of the pixels in the ground-truth image, and add Gaussian noise with standard deviation $0.1$.
4. **Colorization**: We average the color channels in the ground-truth image (with a weight of $\frac{1}{3}$ for each color channel), and add Gaussian noise with standard deviation $0.25$.

**ImageNet restoration.** For the general-content (ImageNet) image restoration tasks in the controlled experiments, we consider the following degradations:

1. **Denoising**: We apply additive white Gaussian noise with standard deviation $0.2$.
2. **Super-resolution**: We use the $4\times$ bicubic down-sampling operator, and add Gaussian noise with standard deviation $0.05$.
3. **Colorization**: We average the color channels in the ground-truth image (with a weight of $\frac{1}{3}$ for each color channel), and add Gaussian noise with standard deviation $0.05$.

### C.2 Implementation details of the flow methods

#### C.2.1 Training

**Face restoration.** For all the face restoration tasks in Section 5.2, the models are trained on the FFHQ data set with images of size $256 \times 256$ (we down-sample the original $1024 \times 1024$ images to $256 \times 256$). Unlike in the blind face image restoration experiments, where the model is trained on images of size $512 \times 512$, here we choose to use a smaller image resolution to save computational resources and achieve shorter training times. We use random horizontal flips for data augmentation.

**ImageNet restoration.** The general-content image restoration models are trained on the ImageNet (Deng et al., 2009) training data, after resizing the images to $128 \times 128$ pixels. To obtain these images, we first resize the original images to have a shorter side of 128 pixels, and then perform random cropping to obtain the desired size. We use random horizontal flips for data augmentation.

#### C.2.2 Choice of $\sigma_s$

As expected, we observe that using $\sigma_s = 0$ in both PMRF (Algorithm 1) and the flow from $Y$ method (Algorithm 4) leads to blurry results with small MSE and large FID. Thus, for a fair comparison, we use the same value of $\sigma_s > 0$ in both methods. We use $\sigma_s = 0.025$ for all the ImageNet restoration tasks and for the face image denoising task. For the rest of the face restoration tasks (inpainting, colorization, and super-resolution), we use $\sigma_s = 0.1^5$.

---

[5] Note that the "optimal" value of $\sigma_s$ depends on the severity of the restoration task. For example, in a mild image denoising task, the posterior mean $\hat{X}^*$ may already be close to the ground-truth image, so $\sigma_s$ should be smaller compared to a case where the noise is severe.

### C.2.3 VECTOR FIELD

**Face restoration.** Similarly to Appendix B.1, the vector field is a HDiT model. The time $t$ in Algorithms 1 and 2 to 4 is sampled from $U[0,1]$ using a stratified sampling strategy. For all baseline methods and PMRF, we train the vector field for 1000 epochs, use a fixed batch size of 256, adopt the AdamW optimizer with a learning rate of $5 \times 10^{-4}$, $(\beta_1, \beta_2) = (0.9, 0.95)$, and a weight decay of $10^{-2}$. As in (Crowson et al., 2024), we do not apply learning rate scheduling. Finally, we use the EMA weights for evaluation, using a decay of 0.9999. A summary of the hyper-parameters is provided in Table 12.

**ImageNet restoration.** The vector field remains the exact same HDiT model as in the face restoration experiments. Here, the model is trained for 100 epochs, and the rest of the hyper-parameters (optimization, EMA for evaluation, *etc*.) remain the same as before.

### C.2.4 POSTERIOR MEAN PREDICTOR

**Face restoration.** The posterior mean predictor $f_\omega$ is a 4.4M parameters SwinIR model[6] which we train from scratch for each task. In all tasks, this model is trained for 1000 epochs, with a fixed batch size of 256, using the AdamW optimizer with a learning rate of $5 \times 10^{-4}$, $(\beta_1, \beta_2) = (0.9, 0.95)$, without weight decay, and without learning rate scheduling. When utilizing this model in the flow process (*e.g*., in PMRF), we use the EMA weights computed with a decay of 0.9999.

**ImageNet restoration.** For the posterior mean predictor $f_\omega$, we use the exact same HDiT model as in appendix C.2.3. Namely, for the general-content restoration experiments, the posterior mean predictor and the vector field models are the same. This model is trained for 100 epochs in all tasks. The rest of the training hyper-parameters (optimization, EMA, *etc*.) remain the same as the those of the SwinIR model described above.

### C.2.5 EVALUATION

**Face restoration.** We test all models on the CelebA-Test data set, with images of size $256 \times 256$. We utilize the `torch-fidelity` package (Obukhov et al., 2020) to compute FID, using the default `inception-v3-compat` image feature extractor (Szegedy et al., 2016). The FID of each method is computed between the entire FFHQ $256 \times 256$ training set, and the images produced by the algorithm for the synthesized CelebA-Test degraded images.

**ImageNet restoration.** We test all models on the ImageNet validation data set (50,000 images), with images resized to $128 \times 128$ pixels. To obtain these images, we first resize the original images to have a shorter side of 128 pixels, followed by center cropping to the desired size. We again utilize the `torch-fidelity` package to compute $\text{FD}_{\text{CLIP}}$, which is the Fréchet distance in the latent space of the `clip-vit-b-32` image feature extractor (Radford et al., 2021) (using this model instead of `inception-v3-compat` ensures a better alignment with human opinion scores Stein et al. (2023)). The $\text{FD}_{\text{CLIP}}$ of each method is computed between the entire ImageNet validation data set (ground-truth images) and the images produced by the algorithm for the corresponding synthesized degraded images.

### C.3 DETAILS OF DOT

**Face restoration.** We utilize DOT (Adrai et al., 2023) similarly to Appendix B.3, using images of size $256 \times 256$ instead of $512 \times 512$, and adopting the official codes of the authors. To compute the latent empirical mean and covariance of the target distribution, we provide the first 1000 from FFHQ to the official code of DOT. For the source distribution, we randomly synthesize degraded images according to the degradation of each task (Appendix C.1) from the first 1000 images in FFHQ, reconstruct each image using the SwinIR model we trained for each task (the same weights we use in PMRF), and finally compute the empirical mean and covariance of the reconstructions in latent space.

---

[6]We use the official code for the SwinIR architecture from `https://github.com/JingyunLiang/SwinIR`. Implementation details and hyper-parameters are provided in our code.

**ImageNet restoration.** We select the first image from each class in the ImageNet training data. To compute the latent empirical mean and covariance of the target distribution, we provide the gathered 1000 images to the official code of DOT. For the source distribution, we degrade each of the collected 1000 ground-truth images according to the degradation of each task (described in appendix C.1), reconstruct the results using the trained ImageNet posterior mean predictor model (described in appendix C.2.4), and finally compute the empirical mean and covariance of the reconstructions in latent space.

# D PROVING THAT FLOW FROM $Y$ IS ALSO OPTIMAL IN EXAMPLE 1

In Section 5.2 we show that, for the denoising task, PMRF and flow from $Y$ are on-par in terms of both perceptual quality and MSE. To provide intuition for this result, we show that flow from $Y$ leads to the desired estimator $\hat{X}_0$ in Example 1 (just like PMRF does).

Specifically, as in Example 1, suppose that $X \sim \mathcal{N}(0,1)$, $N \sim \mathcal{N}(0, \sigma_N^2)$, $\sigma_N > 0$, and $Y = X + N$. In flow from $Y$ with $\sigma_s = 0$ we have $Z_t = tX + (1-t)Y$, and thus $v_{\text{RF}}(Z_t, t) = \mathbb{E}[X - Y | Z_t]$. Below, we show that

$$\text{Cov}(X - Y, Z_t) = (t - 1)\sigma_N^2, \quad \text{and} \tag{41}$$

$$\text{Var}(Z_t) = \sigma_N^2(t^2 - 2t + 1) + 1. \tag{42}$$

Hence,

$$v_{\text{RF}}(Z_t, t) = \mathbb{E}[X - Y | Z_t]$$

$$= \mathbb{E}[X - Y] + \frac{\text{Cov}(X - Y, Z_t)}{\text{Var}(Z_t)}(Z_t - \mathbb{E}[Z_t])$$

$$= \frac{\text{Cov}(X - Y, Z_t)}{\text{Var}(Z_t)} Z_t \tag{43}$$

$$= \frac{(t - 1)\sigma_N^2}{\sigma_N^2(t^2 - 2t + 1) + 1} Z_t, \tag{44}$$

where Equation (43) holds since $\mathbb{E}[X - Y] = 0$ and $\mathbb{E}[Z_t] = 0$. One can verify that the solution of $d\hat{Z}_t = v_{\text{RF}}(\hat{Z}_t, t)dt$ for any initial condition $\hat{Z}_0 = c$ is given by

$$\hat{Z}_t = c \frac{\sqrt{\sigma_N^2(t^2 - 2t + 1) + 1}}{\sqrt{1 + \sigma_N^2}}. \tag{45}$$

Namely, we have

$$\hat{Z}_1 = \frac{1}{\sqrt{1 + \sigma_N^2}} Y$$

$$= \hat{X}_0, \tag{46}$$

where the last equality follows from Equation (28). It follows that flow from $Y$ is also optimal in Example 1, just like PMRF.

Demonstrating Equations (41) and (42) is straightforward. We have

$$\text{Cov}(X - Y, Z_t) = \text{Cov}(X - Y, tX + (1-t)Y)$$

$$= t\text{Cov}(X, X) + (1-t)\text{Cov}(X, Y) - t\text{Cov}(X, Y) - (1-t)\text{Cov}(Y, Y)$$

$$= t + (1-t) - t - (1-t)(1 + \sigma_N^2)$$

$$= (t - 1)\sigma_N^2, \tag{47}$$

and

$$
\begin{aligned}
\mathrm{Var}(Z_t) &= \mathrm{Cov}(tX + (1-t)Y, tX + (1-t)Y) \\
&= t^2\mathrm{Cov}(X, X) + 2t(1-t)\mathrm{Cov}(X, Y) + (1-t)^2\mathrm{Cov}(Y, Y) \\
&= t^2 + 2t(1-t) + (1-t)^2(1 + \sigma_N^2) \\
&= t^2 + 2t - 2t^2 + (1 - 2t + t^2)(1 + \sigma_N^2) \\
&= t^2(1 - 2 + 1 + \sigma_N^2) + 2t(1 - 1 - \sigma_N^2) + 1 + \sigma_N^2 \\
&= t^2\sigma_N^2 - 2t\sigma_N^2 + \sigma_N^2 + 1 \\
&= \sigma_N^2(t^2 - 2t + 1) + 1.
\end{aligned}
\tag{48}
$$

## E  INDICATOR RMSE (INDRMSE) DERIVATION

The MSE of any estimator $\hat{X}$ can always be written as

$$
\mathbb{E}[\|X - \hat{X}\|^2] = \mathbb{E}[\|\hat{X} - \hat{X}^*\|^2] + \mathbb{E}[\|X - \hat{X}^*\|^2] \tag{49}
$$

$$
= \mathbb{E}[\|\hat{X} - \hat{X}^*\|^2] + m, \tag{50}
$$

where $\hat{X}^* = \mathbb{E}[X|Y]$ is the MMSE estimator, Equation (49) follows from Lemma 2 in (Freirich et al., 2021) (Appendix B.1), and $m$ is some constant that does not depend on $\hat{X}$. Thus, if $f(Y) \approx \hat{X}^*$, we have

$$
\mathbb{E}[\|X - \hat{X}\|^2] \approx \mathbb{E}[\|\hat{X} - f(Y)\|^2] + m, \tag{51}
$$

so $\sqrt{\mathbb{E}[\|\hat{X} - f(Y)\|^2]}$ may be used as an indicator for $\sqrt{\mathbb{E}[\|X - \hat{X}\|^2]}$. Future works should investigate the effectiveness of this measure.

Table 2: Varying the number of flow steps $K$ in PMRF (Algorithm 1) on the **CelebA-Test** blind face image restoration benchmark. Red, blue and green indicate the best, the second best, and the third best scores, respectively. Increasing the number of steps improves the perceptual quality while hindering the distortion. These results are expected due to the distortion-perception tradeoff.

| $K$ | Perceptual Quality | | | | Distortion | | | | |
|---|---|---|---|---|---|---|---|---|---|
| | FID↓ | KID↓ | NIQE↓ | Precision↑ | PSNR↑ | SSIM↑ | LPIPS↓ | Deg↓ | LMD↓ |
| 3 | 81.81 | 0.0811 | 8.9012 | 0.2820 | 27.668 | 0.7669 | 0.3582 | 31.41 | 2.0340 |
| 5 | 63.77 | 0.0581 | 7.4568 | 0.4563 | 27.498 | 0.7601 | 0.3401 | 30.80 | 2.0294 |
| 10 | 44.39 | 0.0342 | 5.2648 | 0.6427 | 27.017 | 0.7388 | 0.3314 | 30.49 | 2.0215 |
| 25 | 37.46 | 0.0257 | 4.1179 | 0.7073 | 26.373 | 0.7073 | 0.3470 | 30.67 | 2.0303 |
| 50 | 36.63 | 0.0244 | 3.8492 | 0.7050 | 26.028 | 0.6896 | 0.3591 | 30.89 | 2.0409 |
| 100 | 36.57 | 0.0240 | 3.7311 | 0.7010 | 25.810 | 0.6787 | 0.3662 | 31.06 | 2.0409 |

Table 3: Varying the number of flow steps $K$ in PMRF (Algorithm 1) on the **LFW-Test** blind face image restoration benchmark. Red, blue and green indicate the best, the second best, and the third best scores, respectively. Increasing the number of steps generally improves the perceptual quality while hindering the IndRMSE. These results are expected due to the distortion-perception tradeoff.

| $K$ | FID↓ | KID↓ | NIQE↓ | Precision↑ | IndRMSE↓ |
|---|---|---|---|---|---|
| 3 | 78.2331 | 0.0692 | 8.2315 | 0.3477 | 3.3934 |
| 5 | 64.3121 | 0.0524 | 6.8733 | 0.5143 | 3.8008 |
| 10 | 51.9845 | 0.0387 | 4.9896 | 0.6546 | 4.8648 |
| 25 | 49.3151 | 0.0366 | 4.0028 | 0.6692 | 6.1382 |
| 50 | 49.5581 | 0.0375 | 3.7126 | 0.6826 | 6.7960 |
| 100 | 49.6561 | 0.0377 | 3.6242 | 0.6710 | 7.2004 |

Table 4: Varying the number of flow steps $K$ in PMRF (Algorithm 1) on the **WIDER-Test** blind face image restoration benchmark. Red, blue and green indicate the best, the second best, and the third best scores, respectively. Increasing the number of steps generally improves the perceptual quality while hindering the IndRMSE. These results are expected due to the distortion-perception tradeoff.

| $K$ | FID↓ | KID↓ | NIQE↓ | Precision↑ | IndRMSE↓ |
|---|---|---|---|---|---|
| 3 | 85.0361 | 0.0704 | 9.9988 | 0.2742 | 5.3486 |
| 5 | 65.2563 | 0.0451 | 8.4650 | 0.5381 | 5.7665 |
| 10 | 42.5002 | 0.0179 | 5.5677 | 0.7144 | 7.1134 |
| 25 | 41.2685 | 0.0160 | 4.0726 | 0.7144 | 9.2164 |
| 50 | 41.4446 | 0.0174 | 3.6953 | 0.6845 | 10.3403 |
| 100 | 42.9437 | 0.0183 | 3.5704 | 0.6907 | 11.0674 |

Table 5: Varying the number of flow steps $K$ in PMRF (Algorithm 1) on the **WebPhoto-Test** blind face image restoration benchmark. Red, blue and green indicate the best, the second best, and the third best scores, respectively. Increasing the number of steps generally improves the perceptual quality while hindering the IndRMSE. These results are expected due to the distortion-perception tradeoff.

| $K$ | FID↓ | KID↓ | NIQE↓ | Precision↑ | IndRMSE↓ |
|---|---|---|---|---|---|
| 3 | 128.7858 | 0.0996 | 9.1626 | 0.3907 | 3.2961 |
| 5 | 113.4734 | 0.0782 | 7.5893 | 0.5553 | 3.7371 |
| 10 | 91.3677 | 0.0484 | 5.4199 | 0.6413 | 4.8369 |
| 25 | 81.0642 | 0.0347 | 4.2402 | 0.6462 | 6.3098 |
| 50 | 78.7174 | 0.0324 | 3.9512 | 0.6265 | 7.0159 |
| 100 | 79.1239 | 0.0313 | 3.7990 | 0.5602 | 7.6887 |

Table 6: Varying the number of flow steps $K$ in PMRF (Algorithm 1) on the **CelebAdult-Test** blind face image restoration benchmark. Red, blue and green indicate the best, the second best, and the third best scores, respectively. Increasing the number of steps generally improves the perceptual quality while hindering the IndRMSE. These results are expected due to the distortion-perception tradeoff.

| $K$ | FID↓ | KID↓ | NIQE↓ | Precision↑ | IndRMSE↓ |
|---|---|---|---|---|---|
| 3 | 122.8780 | 0.0551 | 6.6818 | 0.3944 | 3.7339 |
| 5 | 113.7837 | 0.0426 | 5.5810 | 0.4444 | 4.3313 |
| 10 | 105.7426 | 0.0319 | 4.4119 | 0.6111 | 5.4908 |
| 25 | 102.8914 | 0.0293 | 3.7367 | 0.5500 | 6.7145 |
| 50 | 102.1454 | 0.0276 | 3.5609 | 0.6278 | 7.3004 |
| 100 | 102.0568 | 0.0279 | 3.4878 | 0.5944 | 7.7286 |

Table 7: Quantitative evaluation of blind face restoration algorithms on the **LFW-Test** data set.

| Method | FID↓ | KID↓ | NIQE↓ | Precision↑ | IndRMSE↓ |
|---|---|---|---|---|---|
| SwinIR ($\approx$ Posterior mean) | 87.34 | 0.0808 | 8.595 | 0.2513 | 0 |
| DOT | 97.09 | 0.0891 | 5.705 | 0.1806 | 26.24 |
| RestoreFormer++ | 50.80 | 0.0386 | 3.911 | 0.6330 | 9.429 |
| RestoreFormer | 49.04 | 0.0355 | 4.168 | 0.6674 | 12.21 |
| CodeFormer | 52.82 | 0.0387 | 4.484 | 0.6756 | 9.534 |
| VQFRv1 | 51.31 | 0.0399 | 3.590 | 0.6014 | 11.26 |
| VQFRv2 | 51.16 | 0.0378 | 3.761 | 0.6154 | 16.15 |
| GFPGAN | 47.59 | 0.0308 | 4.554 | 0.6400 | 9.842 |
| DiffBIR | 40.97 | 0.0234 | 5.738 | 0.5804 | 9.105 |
| DifFace | 46.48 | 0.0329 | 4.024 | 0.7411 | 11.33 |
| BFRffusion | 50.93 | 0.0377 | 4.963 | 0.6850 | 7.210 |
| **PMRF (Ours)** | 49.32 | 0.0366 | 4.003 | 0.6692 | 6.138 |

Table 8: Quantitative evaluation of blind face restoration algorithms on the **WIDER-Test** data set.

| Method | FID↓ | KID↓ | NIQE↓ | Precision↑ | IndRMSE↓ |
|---|---|---|---|---|---|
| SwinIR ($\approx$ Posterior mean) | 91.96 | 0.0780 | 10.16 | 0.1649 | 0 |
| DOT | 82.15 | 0.0618 | 7.633 | 0.4082 | 14.900 |
| RestoreFormer++ | 45.41 | 0.0209 | 3.759 | 0.6505 | 14.466 |
| RestoreFormer | 50.23 | 0.0251 | 3.894 | 0.6505 | 14.200 |
| CodeFormer | 39.27 | 0.0138 | 4.164 | 0.7227 | 12.185 |
| VQFRv1 | 44.21 | 0.0192 | 3.055 | 0.5959 | 17.042 |
| VQFRv2 | 38.70 | 0.0157 | 3.995 | 0.6381 | 16.368 |
| GFPGAN | 41.28 | 0.0182 | 4.450 | 0.7876 | 11.840 |
| DiffBIR | 35.87 | 0.0114 | 5.659 | 0.6361 | 11.106 |
| DifFace | 37.38 | 0.0131 | 4.383 | 0.7856 | 10.418 |
| BFRffusion | 56.82 | 0.0307 | 4.647 | 0.5825 | 11.759 |
| **PMRF (Ours)** | 41.27 | 0.0160 | 4.073 | 0.7144 | 9.2164 |

Table 9: Quantitative evaluation of blind face restoration algorithms on the **WebPhoto-Test** data set.

| Method | FID↓ | KID↓ | NIQE↓ | Precision↑ | IndRMSE↓ |
|---|---|---|---|---|---|
| SwinIR ($\approx$ Posterior mean) | 132.1 | 0.1022 | 9.638 | 0.2383 | 0 |
| DOT | 125.6 | 0.0865 | 7.397 | 0.3071 | 20.69 |
| RestoreFormer++ | 75.60 | 0.0291 | 4.080 | 0.6143 | 18.43 |
| RestoreFormer | 77.80 | 0.0334 | 4.460 | 0.6265 | 11.55 |
| CodeFormer | 84.17 | 0.0406 | 4.709 | 0.6830 | 8.952 |
| VQFRv1 | 75.57 | 0.0312 | 3.608 | 0.5774 | 12.53 |
| VQFRv2 | 83.52 | 0.0411 | 4.620 | 0.5848 | 14.48 |
| GFPGAN | 88.43 | 0.0494 | 4.941 | 0.6781 | 9.240 |
| DiffBIR | 92.82 | 0.0541 | 6.069 | 0.5307 | 9.152 |
| DifFace | 80.05 | 0.0341 | 4.405 | 0.7273 | 10.31 |
| BFRffusion | 84.83 | 0.0388 | 5.612 | 0.5872 | 7.222 |
| **PMRF (Ours)** | 81.06 | 0.0347 | 4.240 | 0.6462 | 6.310 |

Table 10: Quantitative evaluation of blind face restoration algorithms on the **CelebAdult-Test** data set.

| Method | FID↓ | KID↓ | NIQE↓ | Precision↑ | IndRMSE↓ |
|---|---|---|---|---|---|
| SwinIR ($\approx$ Posterior mean) | 143.80 | 0.0811 | 7.477 | 0.4222 | 0 |
| DOT | 208.54 | 0.1634 | 6.018 | 0.0444 | 44.24 |
| RestoreFormer++ | 103.81 | 0.0313 | 4.006 | 0.5167 | 11.43 |
| RestoreFormer | 103.96 | 0.0315 | 4.320 | 0.5556 | 14.97 |
| CodeFormer | 111.62 | 0.0427 | 4.544 | 0.5722 | 10.49 |
| VQFRv1 | 105.59 | 0.0336 | 3.756 | 0.5944 | 11.14 |
| VQFRv2 | 104.72 | 0.0337 | 3.999 | 0.6056 | 18.51 |
| GFPGAN | 109.19 | 0.0395 | 4.423 | 0.5111 | 11.90 |
| DiffBIR | 109.74 | 0.0411 | 5.650 | 0.5000 | 9.853 |
| DifFace | 98.780 | 0.0243 | 3.901 | 0.6833 | 12.66 |
| BFRffusion | 103.06 | 0.0290 | 4.702 | 0.6056 | 8.037 |
| **PMRF (Ours)** | 102.89 | 0.0293 | 3.737 | 0.5500 | 6.715 |

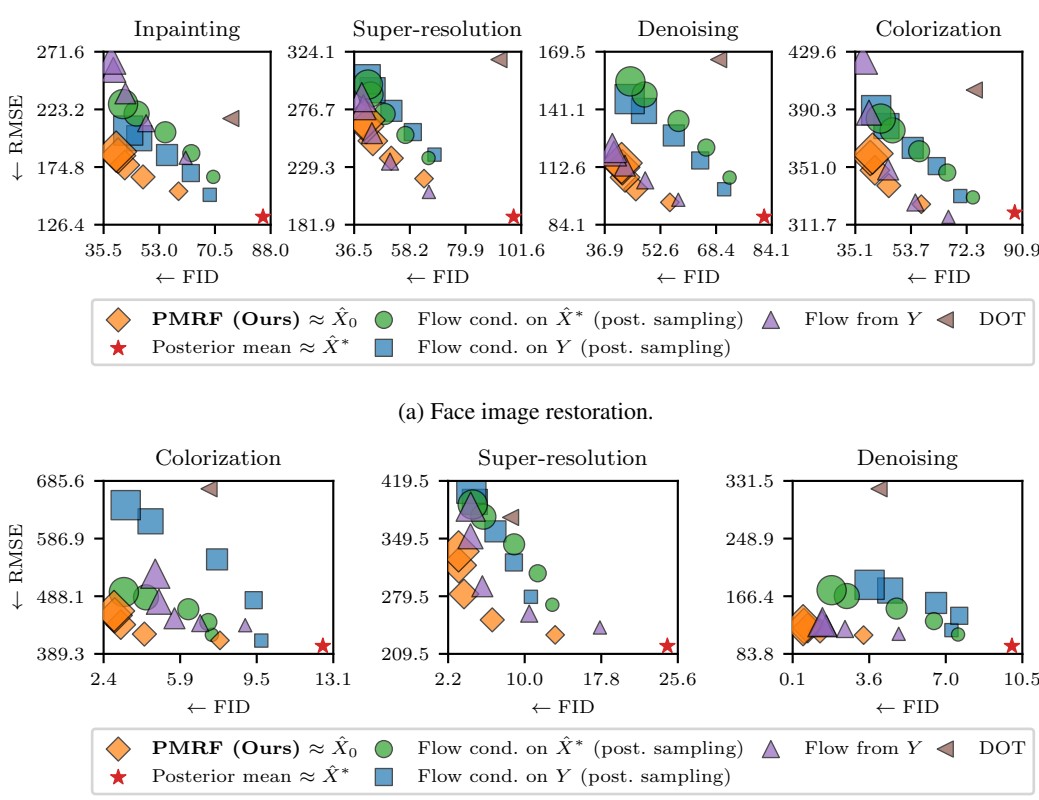

(a) Face image restoration.

(b) ImageNet restoration.

Figure 5: A controlled experiment comparing PMRF with previous methodologies, where we vary the number of steps $K$ in each algorithm (Algorithms 1 and 2 to 4). Specifically, we use $K \in \{5, 10, 20, 50, 100\}$, where a larger marker size corresponds to a larger value of $K$. See Section 5.2 for more details.

Table 11: A comparison of the forward process and training loss of PMRF and the baseline methods from Section 5.2. For the flow from $Y$ algorithm, we have $Y^\dagger = Y$ for all tasks besides super-resolution. For the super-resolution task, we up-scale $Y$ using nearest-neighbor interpolation.

| | **Forward process** | **Flow training loss** |
|---|---|---|
| PMRF (Ours) | $\begin{aligned} Z_t &= tX + (1-t)Z_0 \\ Z_0 &= f_{\omega^*}(Y) + \sigma_s \epsilon \\ \epsilon &\sim \mathcal{N}(0, I) \end{aligned}$ | $\min_\theta \int_0^1 \mathbb{E}\left[\|(X - Z_0) - v_\theta(Z_t, t)\|^2\right] dt$ |
| Flow cond. on $Y$ | $\begin{aligned} Z_t &= tX + (1-t)Z_0 \\ Z_0 &\sim \mathcal{N}(0, I) \end{aligned}$ | $\min_\theta \int_0^1 \mathbb{E}\left[\|(X - Z_0) - v_\theta(Z_t, t, Y)\|^2\right] dt$ |
| Flow cond. on $\hat{X}^*$ | $\begin{aligned} Z_t &= tX + (1-t)Z_0 \\ Z_0 &\sim \mathcal{N}(0, I) \end{aligned}$ | $\min_\theta \int_0^1 \mathbb{E}\left[\|(X - Z_0) - v_\theta(Z_t, t, f_{\omega^*}(Y))\|^2\right] dt$ |
| Flow from $Y$ | $\begin{aligned} Z_t &= tX + (1-t)Z_0 \\ Z_0 &= Y^\dagger + \sigma_s \epsilon \\ \epsilon &\sim \mathcal{N}(0, I) \end{aligned}$ | $\min_\theta \int_0^1 \mathbb{E}\left[\|(X - Z_0) - v_\theta(Z_t, t)\|^2\right] dt$ |

---

**Algorithm 2:** Flow conditioned on $Y$

---

**Training**

Solve $\theta^* \leftarrow \arg\min_\theta \mathbb{E}\left[\|(X - Z_0) - v_\theta(Z_t, t, Y)\|^2\right]$
 `// ` $Z_t \coloneqq tX + (1-t)Z_0$`, ` $Z_0 \sim \mathcal{N}(0, I)$`. ` $t$ `is sampled uniformly from ` $U[0,1]$`.`

**Inference (using Euler's method with $K$ steps to solve the ODE)**

Initialize $\hat{x} \sim \mathcal{N}(0, I)$
**for** $i \leftarrow 0, \ldots, K - 1$ **do**
 $\hat{x} \leftarrow \hat{x} + \frac{1}{K}v_{\theta^*}(\hat{x}, \frac{i}{K}, y)$         `// ` $y$ `is the given degraded measurement`
Return $\hat{x}$

---

**Algorithm 3:** Flow conditioned on $\hat{X}^*$

---

**Training**

*Stage 1:* Solve $\omega^* \leftarrow \arg\min_\omega \mathbb{E}\left[\|X - f_\omega(Y)\|^2\right]$
*Stage 2:* Solve $\theta^* \leftarrow \arg\min_\theta \mathbb{E}\left[\|(X - Z_0) - v_\theta(Z_t, t, f_{\omega^*}(Y))\|^2\right]$
 `// ` $Z_t \coloneqq tX + (1-t)Z_0$`, ` $Z_0 \sim \mathcal{N}(0, I)$`. ` $t$ `is sampled uniformly from ` $U[0,1]$`.`

**Inference (using Euler's method with $K$ steps to solve the ODE)**

Initialize $\hat{x} \sim \mathcal{N}(0, I)$
**for** $i \leftarrow 0, \ldots, K - 1$ **do**
 $\hat{x} \leftarrow \hat{x} + \frac{1}{K}v_{\theta^*}(\hat{x}, \frac{i}{K}, f_{\omega^*}(y))$     `// ` $y$ `is the given degraded measurement`
Return $\hat{x}$

---

**Algorithm 4:** Flow from $Y$

---

**Training**

Solve $\theta^* \leftarrow \arg\min_\theta \mathbb{E}\left[\|(X - Z_0) - v_\theta(Z_t, t)\|^2\right]$         `// ` $Z_t \coloneqq tX + (1-t)Z_0$`,`
 $Z_0 = Y^\dagger + \sigma_s\epsilon$`, ` $\epsilon \sim \mathcal{N}(0, I)$`, and ` $Y^\dagger$ `is the up-scaled version of ` $Y$ `that`
 `matches the dimensionality of ` $X$`. ` $t$ `is sampled uniformly from`
 $U[0,1]$`.`

**Inference (using Euler's method with $K$ steps to solve the ODE)**

Initialize $\hat{x} \sim \mathcal{N}(y^\dagger, I\sigma_s^2)$   `// ` $y^\dagger$ `is the up-scaled version of the degraded`
 `measurement ` $y$
**for** $i \leftarrow 0, \ldots, K - 1$ **do**
 $\hat{x} \leftarrow \hat{x} + \frac{1}{K}v_{\theta^*}(\hat{x}, \frac{i}{K})$
Return $\hat{x}$

---

Table 12: HDiT architecture (Crowson et al., 2024) details and training hyper-parameters.

| Hyper-parameter | Blind face restoration (Section 5.1) | Controlled experiments (Section 5.2) |
|---|---|---|
| Parameters | 160M | 121M |
| GFLOPs / forward | 100.67 | 44.83 (face restoration) |
| | | 11.21 (ImageNet restoration) |
| Memory consumption | 612MB | 464MB |
| Training epochs | 3850 | 1000 (face restoration) |
| | | 100 (ImageNet restoration) |
| Batch size | 256 | 256 |
| Image size | $512{\times}512$ | $256{\times}256$ (face restoration) |
| | | $128{\times}128$ (ImageNet restoration) |
| Precision | bfloat16 mixed | bfloat16 mixed |
| Training hardware | 16 A100 40GB | 4 L40 48GB |
| Training time | 12 days | 2.5 days |
| Patch size | 4 | 4 |
| Levels (local + global attention) | 2 + 1 | 1 + 1 |
| Depth | (2,2,8) | (2,11) |
| Widths | (256,512,1024) | (384,768) |
| Attention heads (width / head dim.) | (4, 8, 16) | (6,12) |
| Attention head dim. | 64 | 64 |
| Neighborhood kernel size | 7 | 7 |
| Mapping depth | 1 | 1 |
| Mapping width | 768 | 768 |
| Optimizer | AdamW | AdamW |
| Learning rate | $5 \cdot 10^{-4}$ | $5 \cdot 10^{-4}$ |
| Learning rate scheduler | Multi-step last 350 epochs | Not applied |
| AdamW betas | (0.9, 0.95) | (0.9, 0.95) |
| AdamW eps. | $10^{-8}$ | $10^{-8}$ |
| Weight decay | $10^{-2}$ | $10^{-2}$ |
| EMA decay | 0.9999 | 0.9999 |

| Ground-truth | Input (synthetic) | DiffBIR | DiffFace | BFRffusion | RestoreFormer++ | GFPGAN | **PMRF (Ours)** |
|---|---|---|---|---|---|---|---|

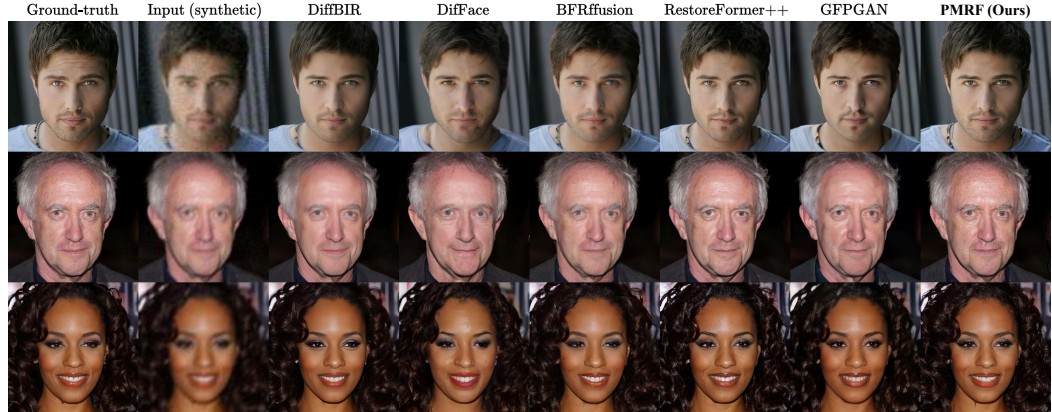

Figure 6: Comparison with state-of-the-art blind face restoration methods on inputs from the **CelebA-Test** data set. Our method produces high perceptual quality while achieving lower distortion overall. **Zoom in for best view**.

| Input (real-world) | DiffBIR | DiffFace | BFRffusion | RestoreFormer++ | GFPGAN | **PMRF (Ours)** |
|---|---|---|---|---|---|---|

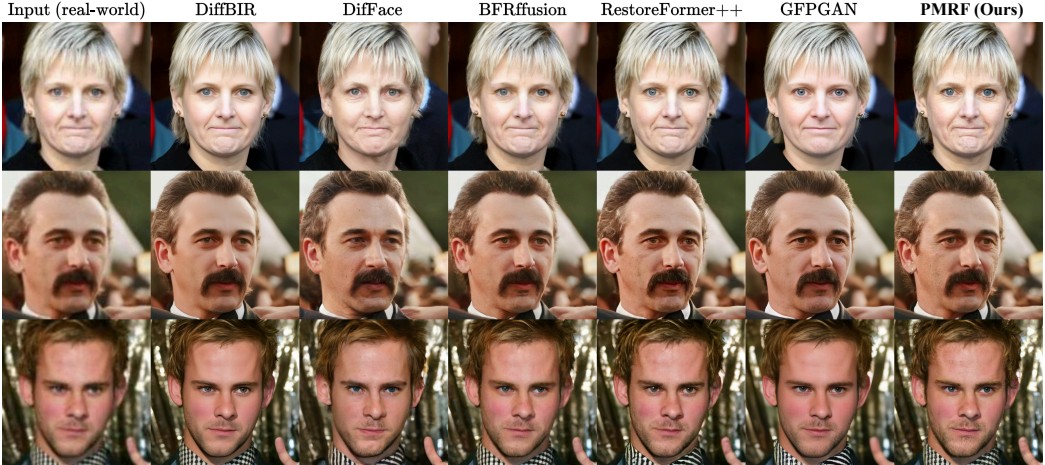

Figure 7: Qualitative results on the real-world **LFW-Test** data set. Our algorithm produces reconstructions with either better or on-par perceptual quality compared to the state-of-the-art, while maintaining very high consistency with the input measurements. **Zoom in for best view**.

| Input (real-world) | DiffBIR | DiffFace | BFRffusion | RestoreFormer++ | GFPGAN | PMRF (Ours) |

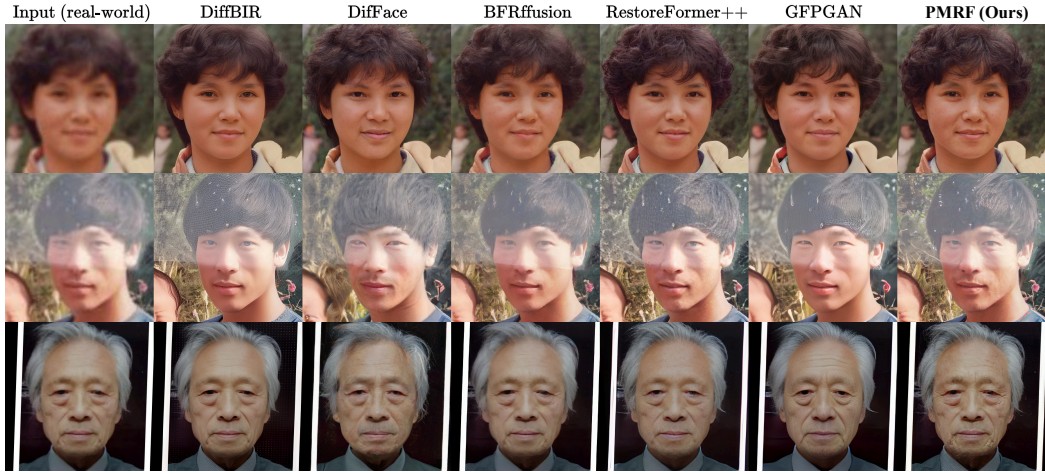

Figure 8: Qualitative results on the real-world **WebPhoto-Test** data set. Our algorithm produces reconstructions with either better or on-par perceptual quality compared to the state-of-the-art, while maintaining very high consistency with the input measurements. **Zoom in for best view**.

| Input (real-world) | DiffBIR | DiffFace | BFRffusion | RestoreFormer++ | GFPGAN | PMRF (Ours) |

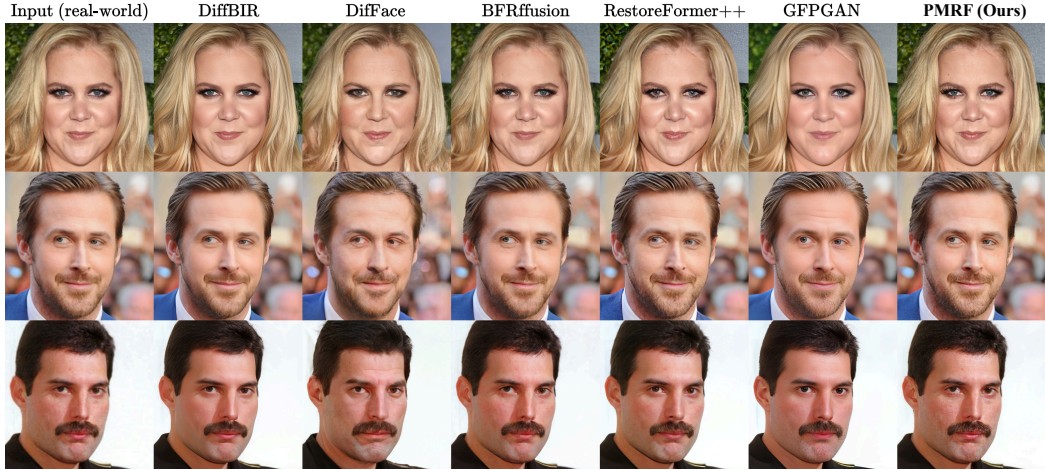

Figure 9: Qualitative results on the real-world **CelebAdult-Test** data set. Our algorithm produces reconstructions with either better or on-par perceptual quality compared to the state-of-the-art, while maintaining very high consistency with the input measurements. **Zoom in for best view**.

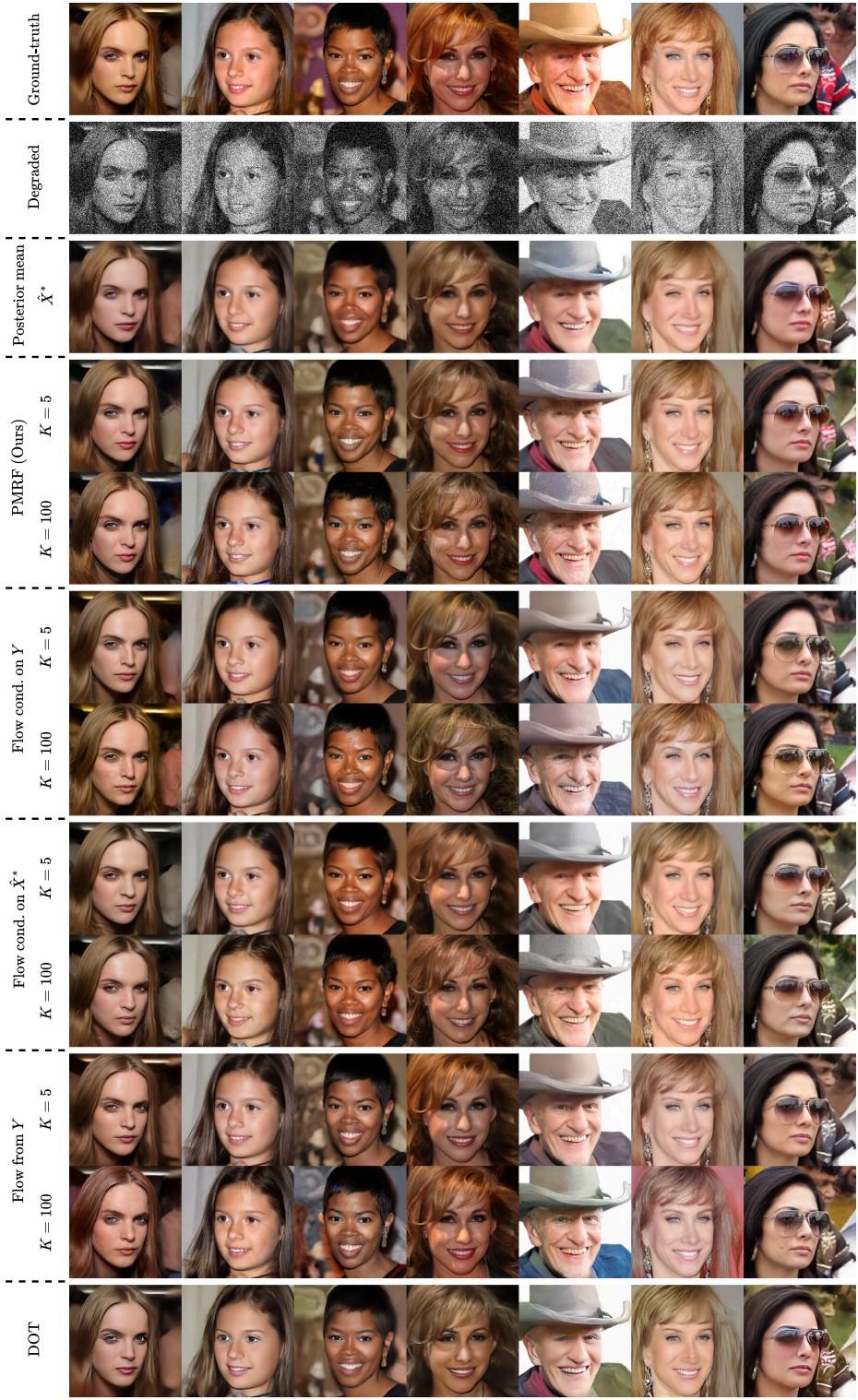

Figure 10: Visual results on the face image **colorization** task from Section 5.2. Our method outperforms all baselines for any number of inference steps $K$. **Zoom in for best view**.

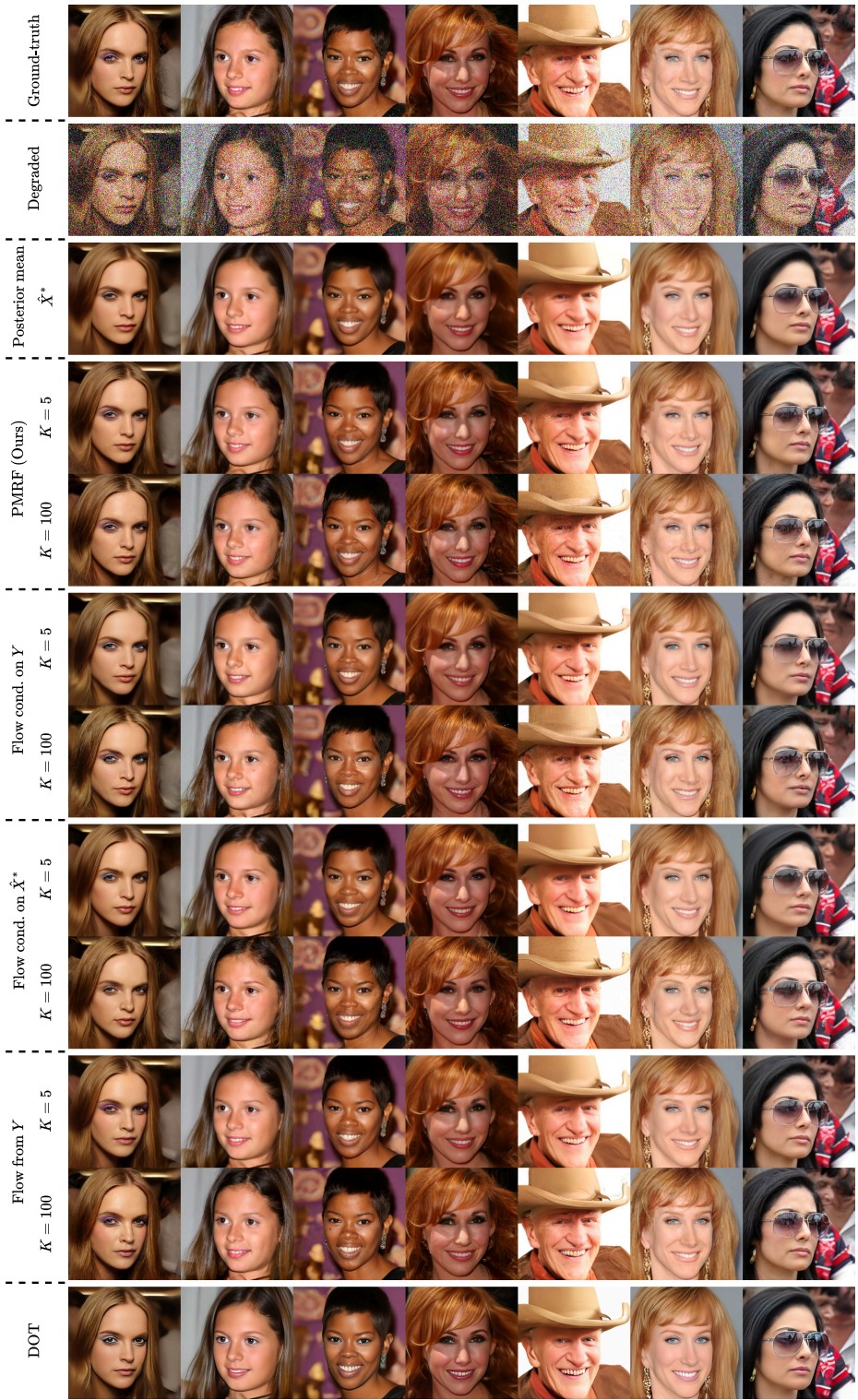

Figure 11: Visual results on the face image **denoising** task from Section 5.2. Our method is on-par with flow from $Y$, and outperforms the posterior sampling methods for any number of inference steps $K$. **Zoom in for best view**.

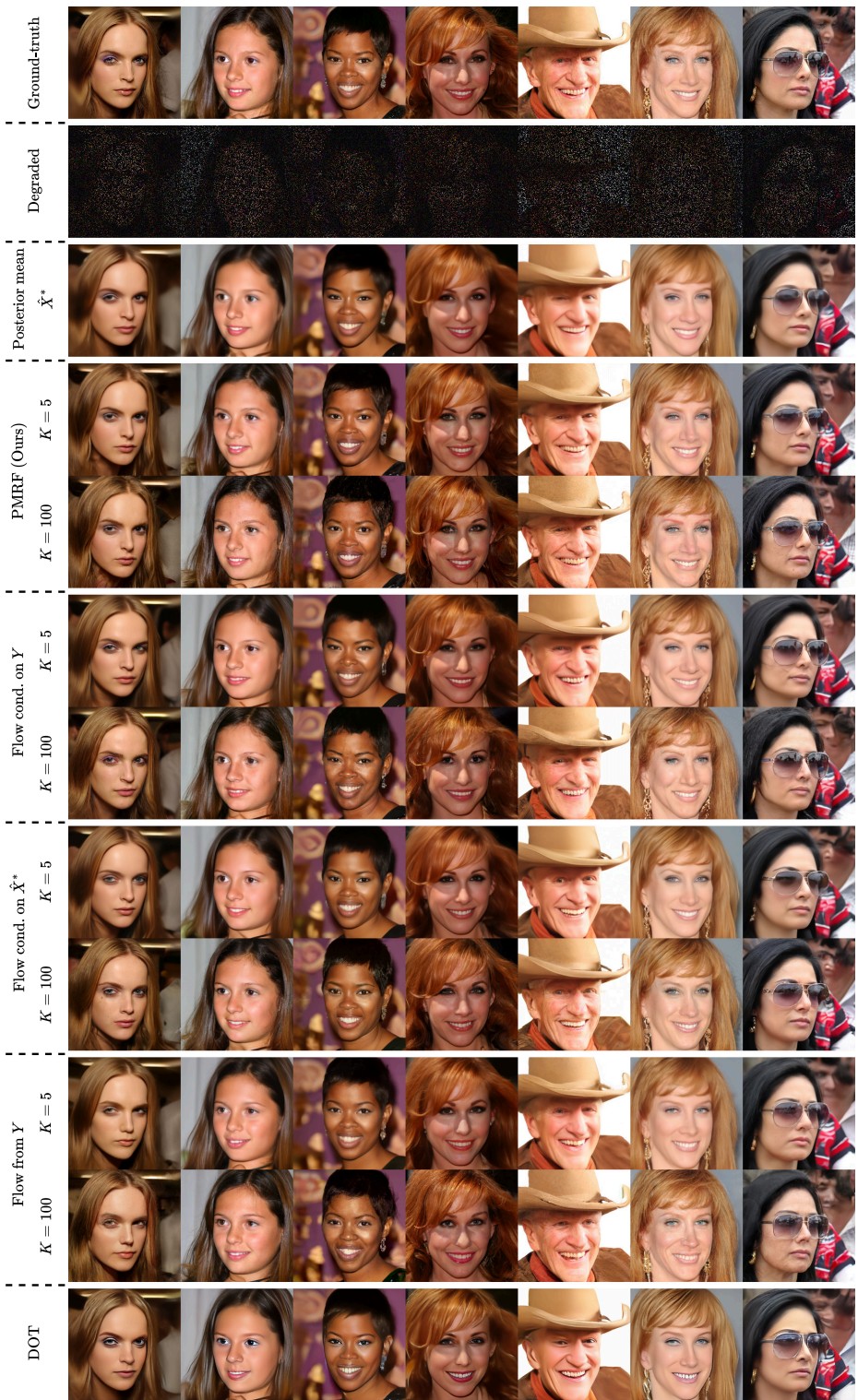

Figure 12: Visual results on the face image **inpainting** task from Section 5.2. Our method outperforms all baselines for any number of inference steps $K$. **Zoom in for best view**.

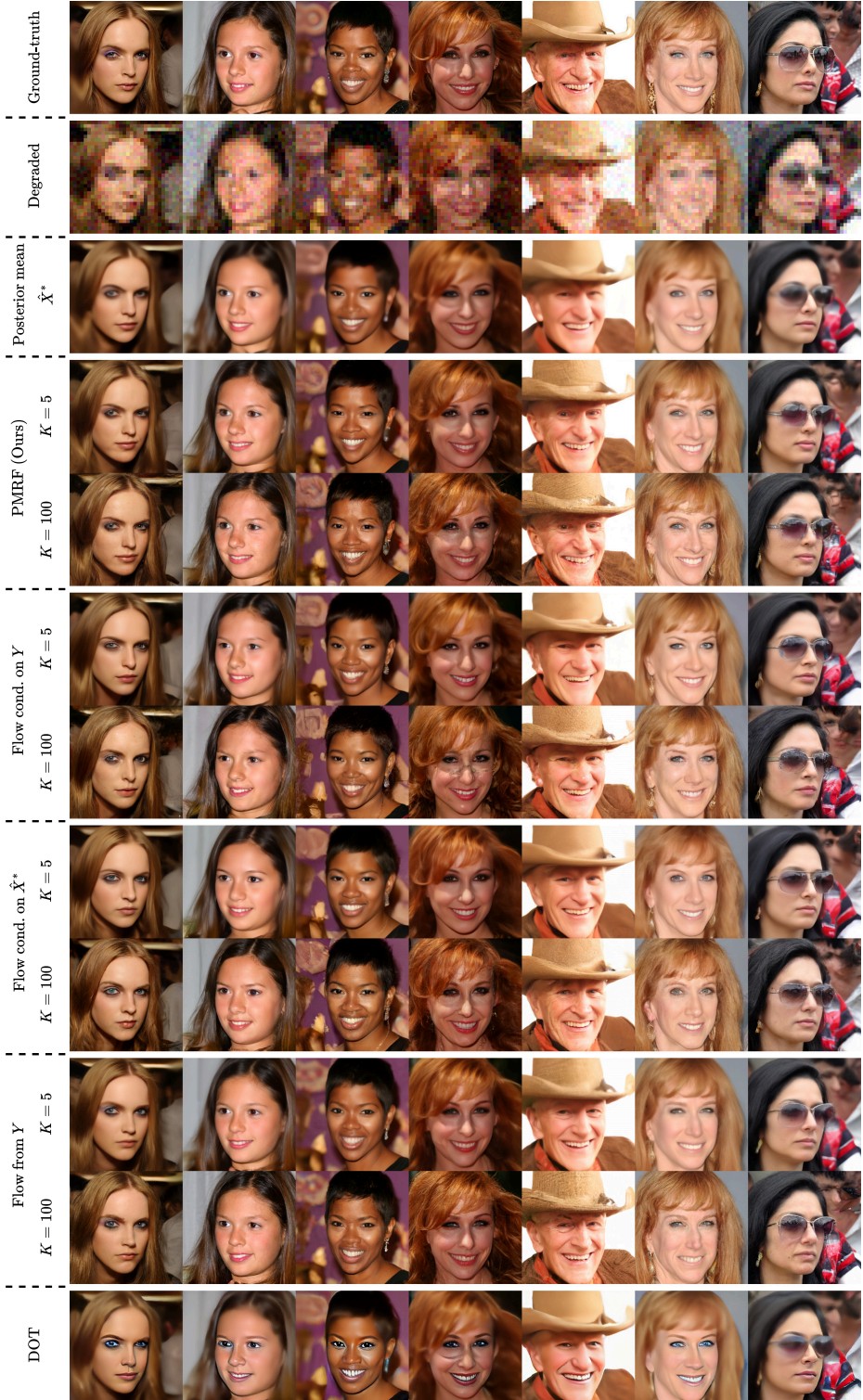

Figure 13: Visual results from Section 5.2 on the face image **super-resolution** task. Our method is on-par with flow from $Y$, and outperforms the posterior sampling methods for any number of inference steps $K$. **Zoom in for best view**.

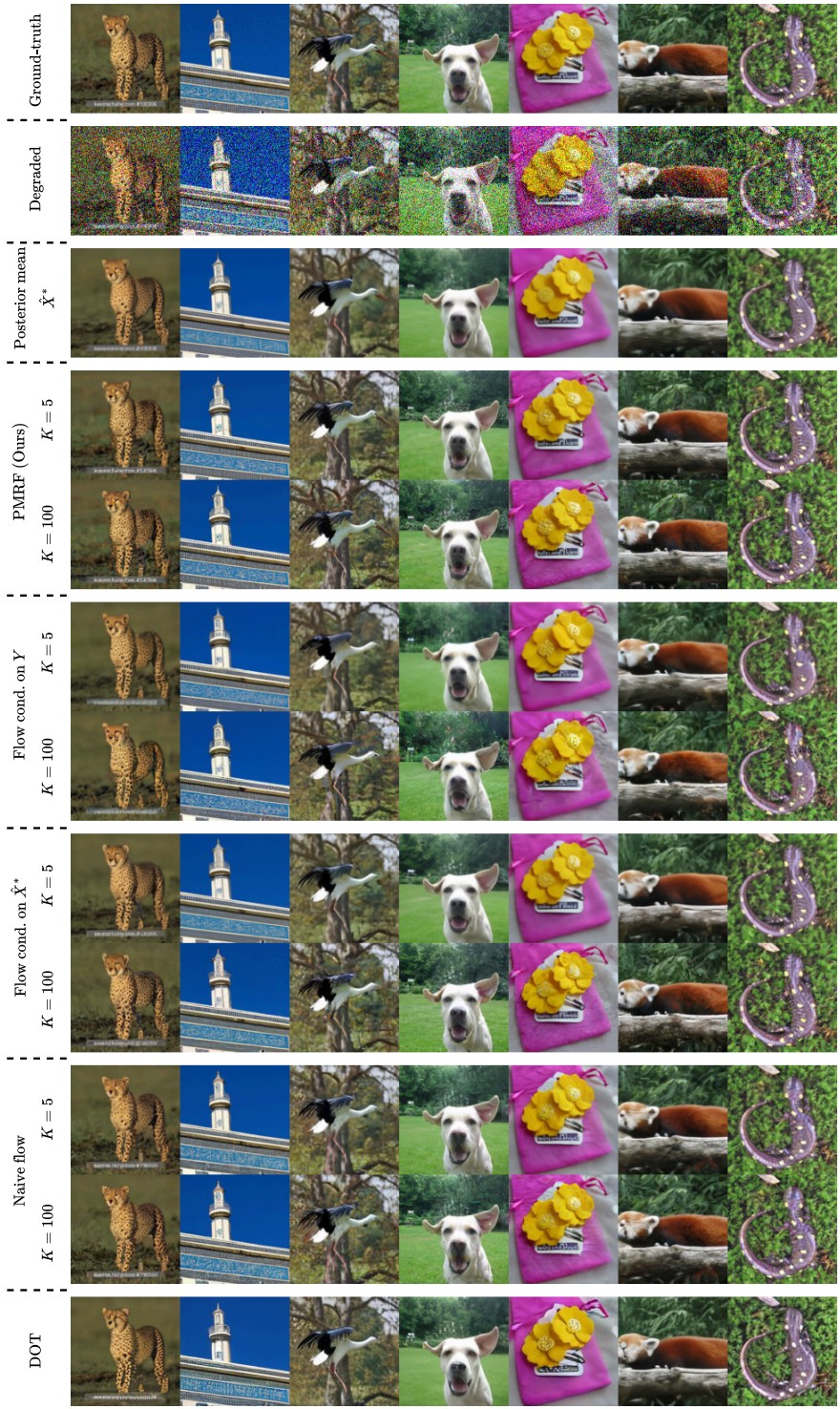

Figure 14: Visual results from Section 5.2 on the ImageNet **denoising** task. Our method outperforms all baselines for any number of inference steps $K$. **Zoom in for best view**.

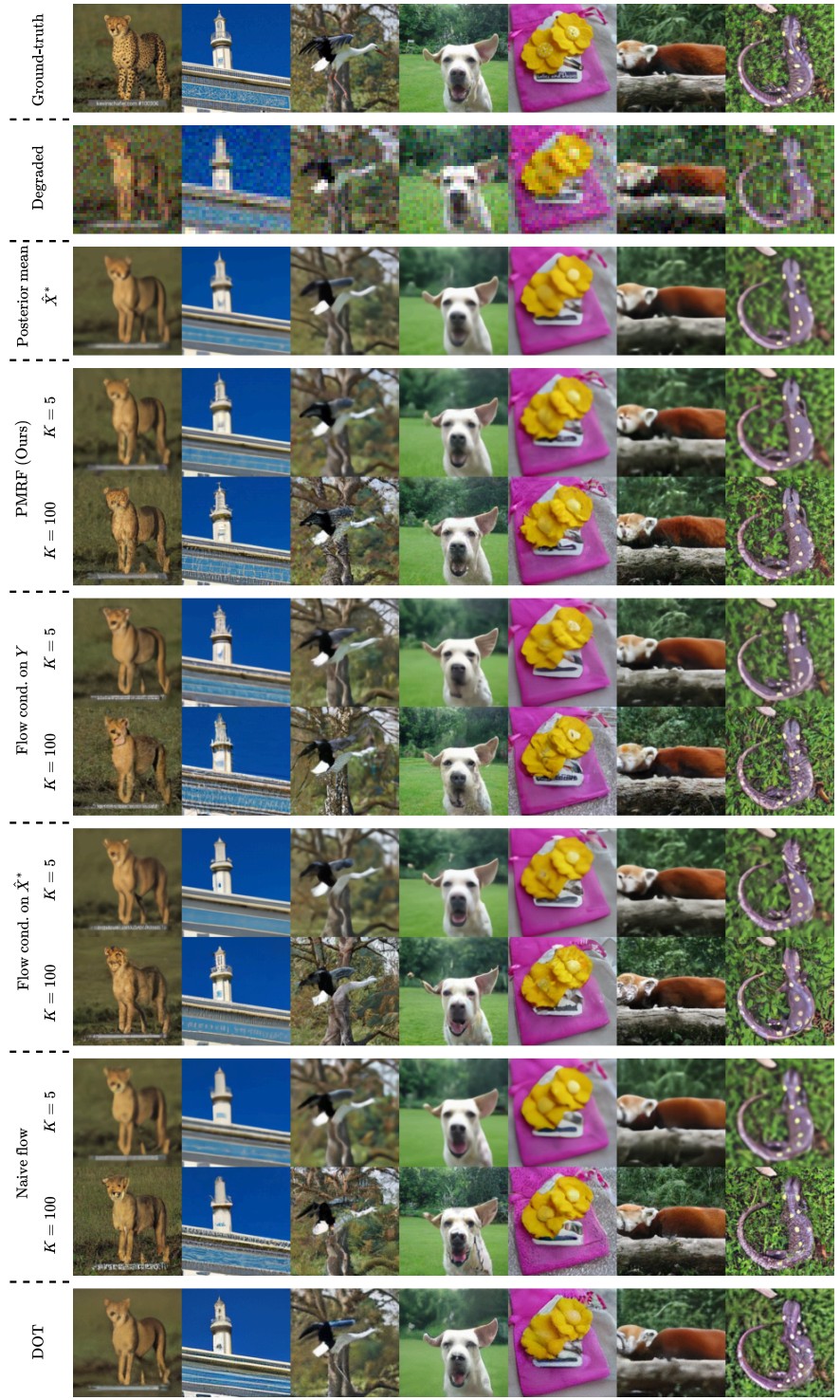

Figure 15: Visual results from Section 5.2 on the ImageNet **super-resolution** task. Our method outperforms all baselines for any number of inference steps $K$. **Zoom in for best view**.

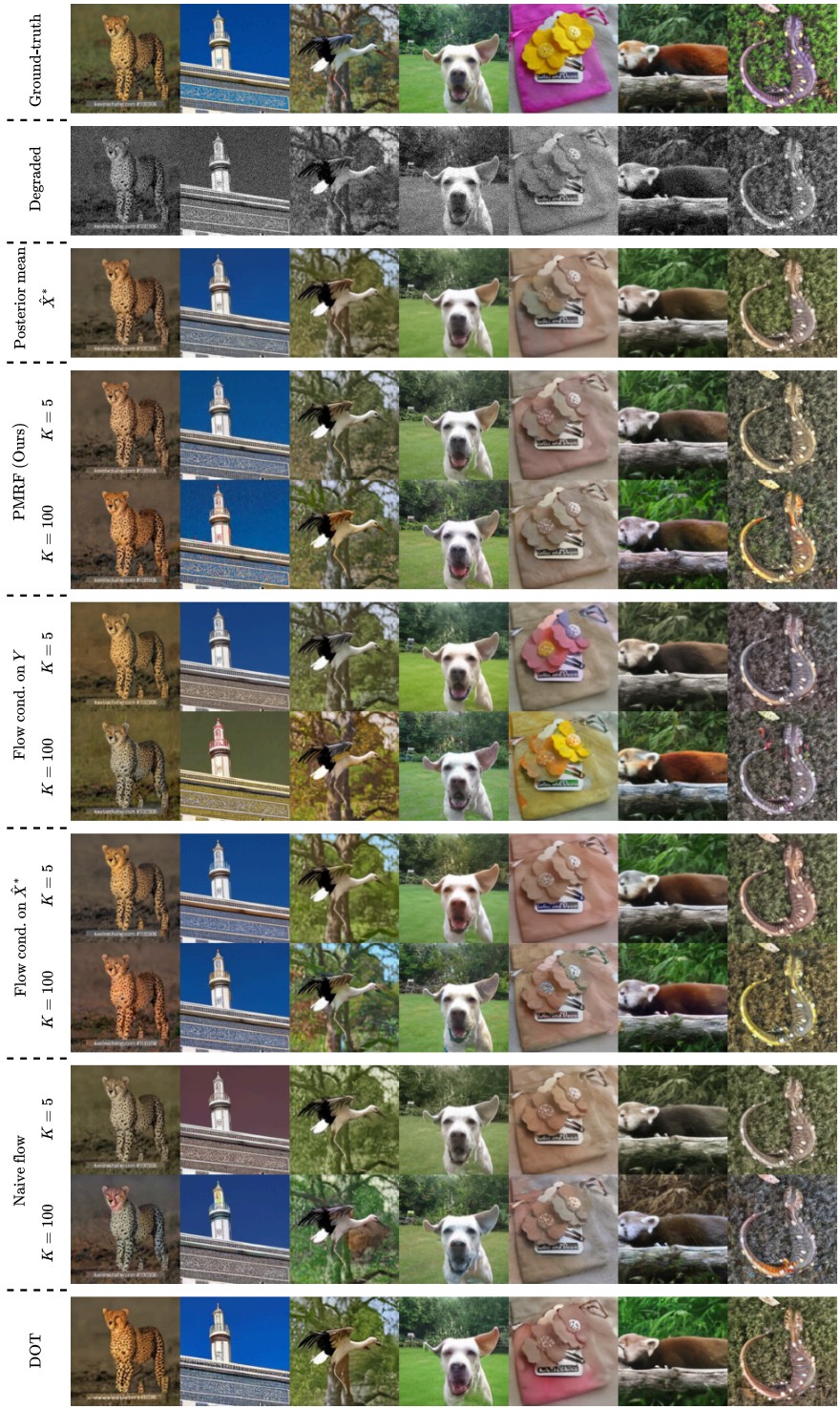

Figure 16: Visual results from Section 5.2 on the ImageNet **colorization** task. Our method outperforms all baselines for any number of inference steps $K$. **Zoom in for best view**.

