# OpenReview forum: "Posterior-Mean Rectified Flow: Towards Minimum MSE Photo-Realistic Image Restoration"
_ICLR.cc/2025/Conference — ICLR 2025 Poster_

### Official Review · Reviewer_rJJP · 2024-10-29

**Soundness:** 4
**Presentation:** 3
**Contribution:** 3
**Rating:** 6
**Confidence:** 4

**Summary:**

The paper introduces posterior-mean rectified flow framework, which approximates optimal estimator under perceptual constraint. The method includes predicting posterior mean and subsequent distribution transport using rectified flow. The experiments on face image restoration tasks validate its efficacy.

**Strengths:**

1.	The motivation is clearly stated. The main idea that considering image restoration problem using MSE optimization under perceptual constraint is interesting and essential.
2.	The experiments on face image restoration demonstrate superior performance over previous methods.

**Weaknesses:**

1. The method seems like such a pipeline: Supervised restoration model trained using MSE loss + SDEdit method (with added $\sigma_s$ noise). Can authors provide further interpretations?
2. The experiments focus on specific the face image domain. The extension to other image domains (such as natural images) is unclear and unexplored.
3. The structure of the paper should be reorganized. The main body does not contain any quantitative results.

**Questions:**

I suggest the authors add experiments on other image domains rather than face domain since the main topic is image restoration problem.

---

> ### Author Response · Authors · 2024-11-21
> **Official response**
>
> We thank the reviewer for their positive feedback on our paper, noting that it is “clear, interesting, and essential.” We address the reviewer’s concerns and questions below.
>
> ## Relation to SDEdit
> Please let us clarify two key distinctions between our PMRF method and SDEdit: (1) SDEdit uses a diffusion model which is trained to transport **white noise** to clean images. In contrast, we use a flow model trained to transport the **posterior mean predictions** (MMSE estimates) to clean images. (2) When relying on a diffusion model that goes from white noise to clean images (as in SDEdit), adding a bit of noise to the image is like going midway into the diffusion process. In our case, it is not the same as going midway into the flow process from MMSE to images: the noise we add is supposed to solve a technical problem, as we explain in Sections 3 and 6.
>
> ## Scope of experiments on the face image domain
> While blind face image restoration remains a highly challenging task, we agree that evaluating PMRF on more diverse datasets (e.g., ImageNet) would better showcase its versatility. Therefore, we have added additional experiments in Section 5.2, where we trained PMRF and the other methods on the ImageNet dataset to solve various tasks: denoising, super-resolution, and colorization. Notably, we observed a consistent trend across both the ImageNet and face restoration tasks: PMRF outperforms all prior frameworks on the distortion-perception plane. We have revised the main text to incorporate these additional experiments (Section 5.2 and Appendix C). We kindly invite you to review our updated results. Thank you for this valuable suggestion!
>
> ## No quantitative results in the main body
> We respectfully clarify that we do provide quantitative results in the main text, for example Table 1, Figure 3, and Figure 4. Figures 3 and 4 illustrate quantitative results on the perception-distortion plane through plots rather than tables, as this visualization offers an intuitive view of the perception-distortion tradeoff. This layout aims to provide readers with a clear understanding of our method’s performance compared to previous methodologies.

---

> > ### Comment · Reviewer_rJJP · 2024-11-23
> > **After Rebuttal**
> >
> > Thank you for the rebuttal. The authors address most of my concerns.

---

### Official Review · Reviewer_amj7 · 2024-10-31

**Soundness:** 3
**Presentation:** 4
**Contribution:** 3
**Rating:** 8
**Confidence:** 3

**Summary:**

The work introduces Posterior-Mean Rectified Flow (PMRF) for image restoration, addressing the challenge of minimizing Mean Squared Error (MSE) while maintaining high perceptual quality. PMRF operates in two stages: first predicting the posterior mean, then using a rectified flow model to transport the result to match the ground-truth image distribution. The authors provide theoretical foundations showing that PMRF can achieve better MSE than posterior sampling methods while maintaining perfect perceptual quality under certain conditions. Through extensive experiments on various image restoration tasks (including blind face restoration, denoising, super-resolution, inpainting, and colorization), they demonstrate that PMRF consistently outperforms or matches state-of-the-art methods in both distortion metrics and perceptual quality measures.

**Strengths:**

+ The paper provides comprehensive experiments and solid theoretical proof, supporting the effectiveness of the proposed method.
+ It is well-written, with clear and informative figures and tables that enhance the reader's understanding.
+ The motivation behind the approach is convincing, and the experimental results substantiate the motivation to a significant extent, demonstrating the practical impact of the proposed framework.

**Weaknesses:**

- While the method shows strong results in blind face image restoration, its evaluation is limited to this domain. Given the theoretical generality of the proposed framework, it would be valuable to see experiments on more diverse and general image datasets to assess the broader applicability of PMRF. Extending their experimental evaluation to include general image restoration would demonstrate the versatility of their method.
- Including ablation studies on the rectified flow model, posterior mean prediction, noise levels, and other architectural choices would provide deeper insights into the importance of each component and help the reader understand which parts of the framework drive the performance improvements.

**Questions:**

If I understand correctly, the idea proposed by the authors should also be effective on general images, beyond the specific tasks tested. It would be interesting to know if the authors have considered conducting experiments or making attempts on general image datasets to further validate the versatility of their method. Of course, the paper is still completely acceptable without these additional experiments. However, including them would significantly enhance the impact of the work.

---

> ### Author Response · Authors · 2024-11-21
> **Official response**
>
> We sincerely thank the reviewer for describing our work as “comprehensive, well-written, and convincing.” We address the noted weaknesses and questions below:
>
> ## Evaluation on general image datasets
> While blind face image restoration remains a highly challenging task, we agree that evaluating PMRF on more diverse datasets (e.g., ImageNet) would better showcase its versatility. Therefore, we have added additional experiments in Section 5.2, where we trained PMRF and the other methods on the ImageNet dataset to solve various tasks: denoising, super-resolution, and colorization.
> Notably, we observed a consistent trend across both the ImageNet and face restoration tasks: PMRF outperforms all prior frameworks on the distortion-perception plane. We have revised the main text to incorporate these additional experiments (Section 5.2 and Appendix C). We kindly invite you to review our updated results. Thank you for this valuable suggestion!
>
> ## Ablation studies on key components
> We agree that ablation studies on components such as the rectified flow model, posterior mean prediction, and noise levels would offer valuable insights. Due to resource constraints and the extensive set of experiments already conducted, we prioritized core evaluations.

---

> > ### Comment · Reviewer_amj7 · 2024-11-26
> >
> > Dear Author,
> >
> > Thank you for your rebuttal and responses to my comments. After reviewing your clarifications and additional information, I find some of them satisfactory. I have decided to maintain my original score.
> >
> > Best regards,
> > Reviewer amj7

---

### Official Review · Reviewer_PKin · 2024-11-03

**Soundness:** 3
**Presentation:** 3
**Contribution:** 3
**Rating:** 6
**Confidence:** 4

**Summary:**

This paper presents a novel approach to photo-realistic image restoration (PIR) that aims to minimize mean squared error (MSE) while maintaining perceptual quality. The authors introduce the Posterior-Mean Rectified Flow (PMRF) algorithm, which optimally transports the posterior mean prediction to the distribution of ground-truth images. This method contrasts with traditional approaches that often prioritize perceptual quality at the expense of distortion. By leveraging a rectified flow model, PMRF effectively approximates the optimal estimator that minimizes MSE under a perfect perceptual index constraint. Experimental results demonstrate that PMRF outperforms existing methods across several restoration tasks.

**Strengths:**

+ PMRF utilizes an optimal transport approach to align the posterior mean with the ground-truth distribution, enhancing the accuracy of image restoration.
+ The paper provides theoretical foundations that demonstrate PMRF’s ability to achieve lower MSE compared to traditional posterior sampling methods.
+ The authors conduct extensive experiments across multiple image restoration tasks, showcasing PMRF’s effectiveness and robustness compared to state-of-the-art techniques.

**Weaknesses:**

- The recent FlowIE work also employs rectified flow for image restoration. I noticed that this paper references FlowIE in the supplementary material. It would be beneficial to include a discussion of this related work and highlight the differences in the main paper.
- In Figures 1 and 4, there are some unnaturally abrupt reflective spots or areas on the forehead, inner eye corners, and cheeks. Are these artifacts caused by the Posterior-Mean Rectified Flow? Please explain the reason behind these effects.

**Questions:**

There are noticeable reflective spots or areas on the forehead, inner eye corners, and cheeks. Are these artifacts a result of the PMRF ? Please clarify the cause of these effects. Additionally, are there any limitations to the proposed method? Could this design be adapted for more general image restoration tasks beyond photo-realistic images?

---

> ### Author Response · Authors · 2024-11-21
> **Official response**
>
> We thank the reviewer for highlighting strengths in our work, including our contributions to "enhancing the accuracy of image restoration, providing theoretical foundations, and conducting extensive experiments across multiple image restoration tasks." We address the reviewer’s points in detail below:
>
> ## Discussing FlowIE in the main text
> Thank you for the great suggestion to highlight the differences between PMRF (our method) and FlowIE in the main text. As we mention in the appendix (L1031), FlowIE is a conditional method that uses a ControlNet (similarly to DiffBIR), which distinguishes it from our PMRF approach. Specifically, FlowIE falls under the category of methods that attempt to sample from the posterior distribution, which differs fundamentally from PMRF’s optimal transport-based approach. We revised the main paper to emphasize this distinction (see L360 in the revised paper). Unfortunately, FlowIE does not currently provide a public checkpoint, so a direct comparison was not possible.
>
> ## Clarification on artifacts
> Regarding the artifacts noted in Figures 1 and 4, we carefully examined the reconstructed images and could not identify specific areas with the reflective spots mentioned (forehead, inner eye corners, and cheeks). If possible, could you share a highlighted example of these artifacts for further clarification? As with most generative models, minor artifacts are not uncommon and may appear in certain reconstructions.
>
> ## Discussing limitations
> Kindly note that we discuss several limitations in Section 6. For instance, our method requires tuning a hyper-parameter $\sigma_s$, which is the standard deviation of the noise added to the posterior mean predictor in the initial PMRF stage. In contrast, posterior sampling methods generally do not require this hyper-parameter adjustment.
>
> ## Could PMRF be adopted for more general image restoration?
> Thank you for pointing out this interesting question. We added new ImageNet experiments to the revised paper (see our new Section 5.2). Importantly, we see that PMRF still dominates the other baseline frameworks on the distortion-perception plane, achieving better distortion with either better or on-par perceptual quality. Namely, these experiments demonstrate the applicability of PMRF to general-content images.
>
> While our current experiments are limited to the natural image domain, we believe that PMRF can also be adopted for other image modalities, such as medical imaging. We hope to see future research exploring PMRF’s potential in such domains.

---

> > ### Comment · Reviewer_PKin · 2024-12-01
> >
> > Dear Authors,
> >
> > Thanks for your detailed response and most of my concerns have been addressed by the response. and I will maintain my score.
> >
> > Best,
> >
> > Reviewer PKin

---

### Official Review · Reviewer_Zdeq · 2024-11-04

**Soundness:** 3
**Presentation:** 3
**Contribution:** 3
**Rating:** 8
**Confidence:** 4

**Summary:**

This paper starts from the distortion-perception tradeoff in image restoration. It first generates the posterior mean and then transports the result to a high quality image following the ground-truth image distoration. The first stage is done by a traditional end-to-end model SwinIR trained with L1 loss, while the second stage is implemented with a rectified flow model by approximating the desired optimal transport map.

**Strengths:**

1, It minimizes the MSE under a perfect perceptual index constraint and achieves a smaller MSE (higher fidelity) than posterior sampling methods (e.g., diffusion-based methods directly restoraing the image from noises).

2, With such a two-stage model, it could be easy-to-train (training a L1 loss-based model is stable) and fast-to-inference (both of the two stages are few-step models).

3, This paper is well-motived, well-written, well-supported and easy-to-understand.

**Weaknesses:**

1, What’s the performance of the baseline model trained the L1 loss? Will it achieve the best PSNR?

2, In addition to distortion metrics, could you please the model’s ability for human face identity perservation?

3, What are the inference speed, memory consumption and FLOPs?

4， The core idea of generating a L1 loss result and then transport it to visually-pleasing image is somewhat similar to the idea of [1] (using the normalizing flow to generate high-quality images conditional the low-quality input). What are their relations?

[1] Hierarchical Conditional Flow: A Unified Framework for Image Super-Resolution and Image Rescaling

**Questions:**

See weakness

---

> ### Author Response · Authors · 2024-11-21
> **Official response**
>
> We would like to thank the reviewer for recognizing our work as "well-motivated, well-written, well-supported, and easy-to-understand." Below, we address the reviewer's questions and points for improvement:
>
> ## Posterior mean predictor model performance
> Kindly note that we report the performance of the baseline models (posterior mean predictors) in Figure 4 in the main text (the red-star ``posterior-mean'' in the controlled experiments) and in Tables 7-10 in the appendix (blind image restoration task). As the reviewer anticipates, these models achieve the best MSE (equivalently, PSNR) in all experiments. However, due to the perception-distortion tradeoff, they exhibit lower perceptual quality. Moreover, we would like to clarify that these models are trained to minimize the MSE (L2) loss, as our goal in the first stage of PMRF is to predict the posterior mean $\mathbb{E}[X|Y]$, which minimizes the MSE. While training a model with the L1 loss (Mean Absolute Error) often achieves similarly high PSNR, we selected the L2 loss to align with our theoretical framework and objective.
>
> ## Identity preservation metric
> Please note that we report the identity preservation metric for the blind face image restoration task in Table 1, denoted as "Deg" - this refers to the embedding angle of ArcFace.  Our model achieves the second-best score in identity preservation, demonstrating competitive performance in maintaining face identity compared to previous methods.
>
> ## Inference Speed, Memory Consumption, and FLOPs.
> We report below the inference speed, GFLOPs and memory consumption of the models from all experiments:
> ### Blind image restoration
> 1. Inference speed/forward pass: 33 milliseconds for the vector field (HDiT model), 38 milliseconds for the posterior mean predictor (SwinIR model). The results are averaged over 1000 forward passes.
> 2. GFLOPs/forward pass: 100.67 for the vector field (HDiT model), 86.8 for the posterior mean predictor (SwinIR model).
> 3. Memory consumption: 612MB for the vector field, 60MB for the posterior mean predictor.
> ### Face image restoration experiments in Section 5.2
> 1. Inference speed/forward pass: 20 milliseconds for the vector field (HDiT model), 33 milliseconds for the posterior mean predictor (SwinIR model). The results are averaged over 1000 forward passes.
> 2. GFLOPs/forward pass: 44.83 for the vector field (HDiT model), 10.4 for the posterior mean predictor (SwinIR model).
> 3. Memory consumption: 463MB for the vector field, 18MB for the posterior mean predictor.
> ### ImageNet restoration experiments in Section 5.2 (newly added for the rebuttal - see the revised paper pdf)
> 1. Inference speed/forward pass: 19 milliseconds for the vector field (HDiT model), 19 milliseconds for the posterior mean predictor (HDiT model). The results are averaged over 1000 forward passes.
> 2. GFLOPs/forward pass: 11.21 for the vector field (HDiT model), 11.21 for the posterior mean predictor (HDiT model).
> 3. Memory consumption: 463MB for the vector field, 463MB for the posterior mean predictor.
>
> We added these details to the appendix (see Table 12 in the paper revision). Thanks for the suggestion!
>
> ## Relation to Hierarchical Conditional Flow (HCF)
> As we understand, HCF is a conditional normalizing flow model trained to generate samples from the posterior distribution. While HCF uses an L1 regularization term to center its outputs around the posterior mean (as noted by the authors of HCF, after Equation 7 in their paper), our approach differs fundamentally by focusing on directly transporting the posterior mean to the target distribution, rather than sampling around it. This superficial difference leads to the compelling theoretical and practical results of our method.

---

### Author Response · Authors · 2024-11-21
**Summary of paper revision**

We sincerely thank the reviewers for their efforts to review our paper and provide valuable feedback. In response to the reviews, we have added new experiments on the ImageNet dataset (discussed in Section 5.2). To facilitate your review of the revised manuscript, we summarize the changes below:

1. **Section 5.1.1**: We now clarify the difference between PMRF (our method) and FlowIE in L360.
2. **Section 5.2**: We have slightly revised this section to include details about the new ImageNet experiments.
3. **Figure 4**: The primary quantitative results of the new ImageNet experiments are now presented alongside the results of the face image restoration experiments, where we also include visual examples. Additionally, **Figure 5 in the appendix** extends these results by evaluating the algorithms with varying numbers of flow inference steps (as we did before for the face image restoration experiments).
4. **Appendix C**: Comprehensive details of the ImageNet experiments have been added.
5. **Table 12 in the appendix**: We updated this table to reflect the training hyper-parameters used in the new experiments, and included additional information such as memory consumption and GFLOPs per forward pass.
6. **New Figures 14–16 in the appendix**: These figures include additional visual examples of the ImageNet experiments.

We truly hope this summary of changes will streamline your review process and facilitate your evaluation of the updates.

---

### Meta-Review · Area_Chair_ftef · 2024-12-20

**Metareview:**

The reviewers found the work with good theoretical background and validation. Author rebuttal addressed most of their concerns. Reviewers rJJP and PKin were also satisfied with the author responses but did not change the original score. Given that all reviewers were positive and convinced after the rebuttal, the paper is recommended for acceptance.

**Additional Comments On Reviewer Discussion:**

Two of the reviewers with lower score were convinced by the authors' response. But they did not raise their initial score. They were not responsive for clarifying this.

---

### Decision · Program_Chairs · 2025-01-22

Accept (Poster)